# Sign and Basis Invariant Networks for Spectral Graph Representation Learning

## Abstract

We introduce SignNet and BasisNet—new neural architectures that are invariant to two key symmetries displayed by eigenvectors: (i) sign flips, since if $v$ is an eigenvector then so is $-v$; and (ii) more general basis symmetries, which occur in higher dimensional eigenspaces with infinitely many choices of basis eigenvectors. We prove that our networks are universal, i.e., they can approximate any continuous function of eigenvectors with the desired invariances. Moreover, when used with Laplacian eigenvectors, our architectures are provably expressive for graph representation learning: they can approximate any spectral graph convolution, can compute spectral invariants that go beyond message passing neural networks, and can provably simulate previously proposed graph positional encodings. Experiments show the strength of our networks for molecular graph regression, learning expressive graph representations, and learning neural fields on triangle meshes.

## 1 Introduction

Numerous machine learning models process eigenvectors, which arise in various scenarios including principal component analysis, matrix factorizations, and operators associated to graphs or manifolds. An important example is the use of Laplacian eigenvectors to encode information about the structure of a graph or manifold [Belkin and Niyogi, 2003, Von Luxburg, 2007, Lévy, 2006]. Positional encodings that involve Laplacian eigenvectors have recently been used to generalize Transformers to graphs [Kreuzer et al., 2021, Dwivedi and Bresson, 2021], and to improve the expressive power and empirical performance of graph neural networks (GNNs) [Dwivedi et al., 2022]. Furthermore, these eigenvectors are crucial for defining spectral operations on graphs that are foundational to graph signal processing and spectral GNNs [Ortega et al., 2018, Bruna et al., 2014].

However, there are nontrivial symmetries that should be accounted for when processing eigenvectors. For instance, if $v$ is an eigenvector, then so is $-v$, with the same eigenvalue. More generally, if an eigenvalue has higher multiplicity, then there are infinitely many unit-norm eigenvectors that can be chosen. Indeed, a full set of orthonormal eigenvectors is only defined up to a change of basis in each eigenspace. In the case of sign invariance, for any $k$ eigenvectors there are $2^k$ possible choices of sign. Accordingly, prior works randomly flip eigenvector signs during training in order to approximately learn sign invariance [Kreuzer et al., 2021, Dwivedi et al., 2020]. However, learning all $2^k$ invariances is challenging and limits the effectiveness of Laplacian eigenvectors for encoding positional information. Sign invariance is a special case of basis invariance when all eigenvalues are distinct, but general basis invariance is even more difficult to deal with. In Appendix C.2, we show that higher dimensional eigenspaces are abundant in real datasets; for instance, 64% of molecule graphs in the ZINC dataset have a higher dimensional eigenspace.

In this work, we address the sign and basis ambiguity problems by developing new neural networks—SignNet and BasisNet. Our networks are universal and can approximate any continuous function

of eigenvectors with the proper invariances. Moreover, our networks are theoretically powerful for graph representation learning—they can approximate spectral graph convolutions and compute powerful spectral invariants, which allows our networks to express graph properties like subgraph counts that message passing neural networks cannot. Finally, Laplacian eigenvectors with SignNet and BasisNet can approximate many previously proposed graph positional encodings, including those based on random walks [Li et al., 2020, Dwivedi et al., 2022] and heat kernels [Mialon et al., 2021, Feldman et al., 2022]. Experiments on molecular graph regression tasks, learning expressive graph representations, and texture reconstruction on triangle meshes illustrate the empirical benefits of our models' approximation power and invariances.

## 2 Sign and Basis Invariant Networks

For an $n \times n$ symmetric matrix, let $\lambda_1 \leq \ldots \leq \lambda_n$ be the eigenvalues and $v_1, \ldots, v_n$ the corresponding eigenvectors, which we may assume to form an orthonormal basis. For instance, we could consider the normalized graph Laplacian $L = I - D^{-1/2}AD^{-1/2}$, where $A \in \mathbb{R}^{n \times n}$ is the adjacency matrix and $D$ is the diagonal degree matrix of some underlying graph. For undirected graphs, $L$ is symmetric. Nonsymmetric matrices can be handled very similarly, as we show in Appendix B.1. Our goal is to parameterize a class of models $f(v_1, \ldots, v_k)$ taking $k$ eigenvectors as input in a manner that respects the eigenvector symmetries.

Figure 1: Symmetries of eigenvectors of a symmetric matrix with permutation symmetries (e.g. a graph Laplacian). A neural network applied to the eigenvector matrix (middle) should be invariant or equivariant to permutation of the rows (left product with a permutation matrix $P$) and invariant to the choice of eigenvectors in each eigenbasis (right product with a block diagonal orthogonal matrix $\mathrm{Diag}(Q_1, Q_2, Q_3)$).

**Sign invariance.** For any of the $v_i$, the sign flipped $-v_i$ is also an eigenvector, so a function $f : \mathbb{R}^{n \times k} \to \mathbb{R}^s$ (where $s$ is an arbitrary output dimension) should be *sign invariant*:

$$f(v_1, \ldots, v_k) = f(s_1 v_1, \ldots, s_k v_k) \qquad (1)$$

for all sign choices $s_i \in \{-1, 1\}$. That is, we want $f$ to be invariant to the product group $\{-1, 1\}^k$. This captures all eigenvector symmetries if the eigenvalues $\lambda_i$ are distinct.

**Basis invariance.** If the eigenvalues have higher multiplicity, then there are further symmetries. Let $V_1, \ldots, V_l$ be bases of eigenspaces—i.e., $V_i = \begin{bmatrix} v_{i_1} & \ldots & v_{i_{d_i}} \end{bmatrix} \in \mathbb{R}^{n \times d_i}$ has orthonormal columns and spans the eigenspace associated with the shared eigenvalue $\mu_i = \lambda_{i_1} = \ldots = \lambda_{i_{d_i}}$. Any other orthonormal basis that spans the eigenspace is of the form $V_i Q$ for some orthogonal $Q \in O(d_i) \subseteq \mathbb{R}^{d_i \times d_i}$ (see Appendix F.2). Thus, a function $f : \mathbb{R}^{n \times \sum_{i=1}^l d_i} \to \mathbb{R}^s$ that is invariant to changes of basis in each eigenspace satisfies

$$f(V_1, \ldots, V_l) = f(V_1 Q_1, \ldots, V_l Q_l), \qquad Q_i \in O(d_i). \qquad (2)$$

In other words, $f$ is invariant to the product group $O(d_1) \times \ldots \times O(d_l)$. The number of eigenspaces $l$ and the dimensions $d_i$ may vary between matrices; we account for this in Section 2.2. As $O(1) = \{-1, 1\}$, sign invariance is a special case of basis invariance when all eigenvalues are distinct.

**Permutation equivariance.** For GNN models that output node features or node predictions, one typically further desires $f$ to be invariant or equivariant to permutations of nodes, i.e., along the entries (or rows) of each vector. Thus, for $f : \mathbb{R}^{n \times d} \to \mathbb{R}^{n \times d}$, we typically also require $f(PV_1, \ldots, PV_l) = Pf(V_1, \ldots, V_l)$ for any permutation matrix $P \in \mathbb{R}^{n \times n}$. Figure 1 illustrates the full setup.

**Graph Positional Encodings.** A major motivation for processing eigenvector input is for graph positional encodings, which are additional features appended to each node in a graph that give information about the position of that node in the graph. These additional features are crucial for generalizing Transformers to graphs, and also have been found to improve performance of GNNs. Figure 2 illustrates a standard pipeline and the use of our SignNet within it: the input adjacency, node features, and eigenvectors of a graph are used to compute a prediction about the graph. Laplacian

eigenvectors are processed before being fed into this prediction model. Laplacian eigenvectors have been widely used as positional encodings, and many works have noted that sign and/or basis invariance must be dealt with in this case [Dwivedi and Bresson, 2021, Beaini et al., 2021, Dwivedi et al., 2020, Kreuzer et al., 2021, Mialon et al., 2021, Dwivedi et al., 2022].

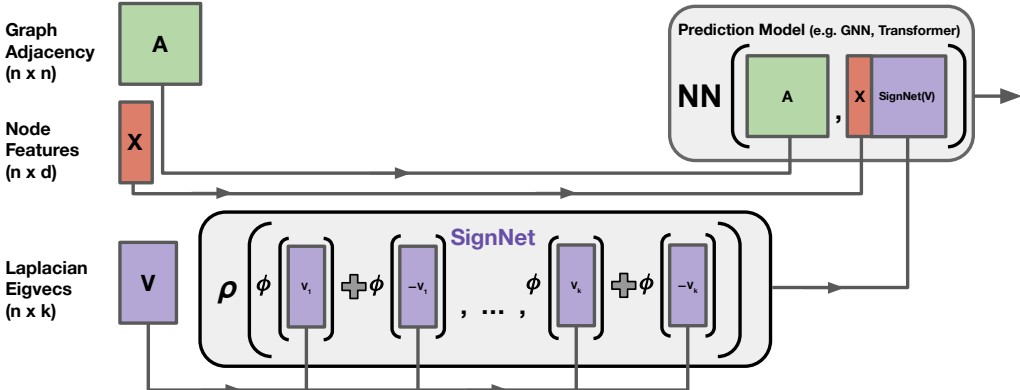

Figure 2: Pipeline for using node positional encodings. After processing by our SignNet, the learned positional encodings from the Laplacian eigenvectors are added as additional node features of an input graph. These positional encodings along with the graph adjacency and original node features are passed to a prediction model (e.g. a GNN). Not shown here, SignNet can also take in eigenvalues and node features if desired.

## 2.1 Warmup: Neural Networks on One Eigenspace

Before considering the general setting, we design neural networks that take a single eigenvector or eigenspace as input and are sign or basis invariant. These single subspace architectures will become building blocks for the general architectures. For one subspace, a sign invariant function is merely an even function, and is easily parameterized.

**Proposition 1.** *A continuous function $h : \mathbb{R}^n \to \mathbb{R}^s$ is sign invariant if and only if*

$$h(v) = \phi(v) + \phi(-v) \tag{3}$$

*for some continuous $\phi : \mathbb{R}^n \to \mathbb{R}^s$. A continuous $h : \mathbb{R}^n \to \mathbb{R}^n$ is sign invariant and permutation equivariant if and only if* (3) *holds for a continuous permutation equivariant $\phi : \mathbb{R}^n \to \mathbb{R}^n$.*

In practice, we parameterize $\phi$ by a neural network. Any architecture choice will ensure sign invariance, while permutation equivariance can be achieved using elementwise MLPs (Multi-Layer Perceptrons), DeepSets [Zaheer et al., 2017], Transformers [Vaswani et al., 2017], or GNNs.

Next, we address basis invariance for a single $d$-dimensional subspace, i.e., we aim to parameterize maps $h : \mathbb{R}^{n \times d} \to \mathbb{R}^n$ that are (a) invariant to right multiplication by $Q \in O(d)$, and (b) equivariant to permutations along the row axis. For (a), we use the mapping $V \mapsto VV^\top$ from $V$ to the orthogonal projector of its column space, which is $O(d)$ invariant. Mapping $V \mapsto VV^\top$ does not lose information if we treat $V$ as equivalent to $VQ$ for any $Q \in O(d)$. This is justified by the classical first fundamental theorem of $O(d)$ [Kraft and Procesi, 1996], which has recently been applied in machine learning by Villar et al. [2021].

Regarding (b), permuting the rows of $V$ permutes rows and columns of $VV^\top \in \mathbb{R}^{n \times n}$. Hence, we desire the function $\phi : \mathbb{R}^{n \times n} \to \mathbb{R}^n$ on $VV^\top$ to be equivariant to both row and column permutation: $\phi(PVV^\top P^\top) = P\phi(VV^\top)$. To parameterize such a mapping from matrices to vectors, we use an invariant graph network (IGN) [Maron et al., 2018]—a neural network mapping to and from tensors of arbitrary order $\mathbb{R}^{n^{d_1}} \to \mathbb{R}^{n^{d_2}}$ that has the desired permutation equivariance. We thus parameterize a family with the requisite invariance and equivariance as follows:

$$h(V) = \text{IGN}(VV^\top). \tag{4}$$

Proposition 2 states that this architecture universally approximates $O(d)$ invariant and permutation equivariant functions. The full approximation power requires high order tensors to be used for the IGN; in practice, we restrict the tensor dimensions for efficiency, as discussed in the next section.

**Proposition 2.** *Any continuous, $O(d)$ invariant $h : \mathbb{R}^{n \times d} \to \mathbb{R}^s$ is of the form $h(V) = \phi(VV^\top)$ for a continuous $\phi$. For a compact domain $\mathcal{Z} \subseteq \mathbb{R}^{n \times d}$, maps of the form $V \mapsto \mathrm{IGN}(VV^\top)$ universally approximate continuous $h : \mathcal{Z} \subseteq \mathbb{R}^{n \times d} \to \mathbb{R}^n$ that are $O(d)$ invariant and permutation equivariant.*

## 2.2 Neural Networks on Multiple Eigenspaces

Next, we use the single-eigenspace models $\phi_l(V_l)$ as building blocks for a model on multiple eigenspaces. To do so, we use functions of the form $f(V_1, \ldots, V_l) = \rho(\phi_1(V_1), \ldots, \phi_l(V_l))$, i.e., we first process each eigenspace individually with an invariant network, and then aggregate them via a function $\rho$. This approach is grounded in a general decomposition theorem for product spaces that we prove in Section A.

**SignNet.** We parameterize our sign invariant network $f : \mathbb{R}^{n \times k} \to \mathbb{R}^s$ on eigenvectors $v_1, \ldots, v_k$ as

$$f(v_1, \ldots, v_k) = \rho\left([\phi(v_i) + \phi(-v_i)]_{i=1}^k\right), \tag{5}$$

where $\phi$ and $\rho$ are unrestricted neural networks, and $[\cdot]_i$ denotes concatenation of vectors. The form $\phi(v_i) + \phi(-v_i)$ induces sign invariance for each eigenvector. Since we do not yet impose permutation equivariance here, we term this model *Unconstrained-SignNet*.

To obtain a sign invariant *and* permutation equivariant $f$ that outputs vectors in $\mathbb{R}^{n \times s}$, we restrict $\phi$ and $\rho$ to be permutation equivariant networks from vectors to vectors, such as elementwise MLPs, DeepSets [Zaheer et al., 2017], Transformers [Vaswani et al., 2017], or most standard GNNs. We name this permutation equivariant version *SignNet*. If desired, we can additionally use eigenvalues $\lambda_i$ and node features $X \in \mathbb{R}^{n \times q}$ by adding them as arguments to $\phi$:

$$f(v_1, \ldots, v_k, \lambda_1, \ldots, \lambda_k, X) = \rho\left([\phi(v_i, \lambda_i, X) + \phi(-v_i, \lambda_i, X)]_{i=1}^k\right). \tag{6}$$

**BasisNet.** For basis invariance, let $V_i \in \mathbb{R}^{n \times d_i}$ be an orthonormal basis of a $d_i$ dimensional eigenspace. Then we parameterize our *Unconstrained-BasisNet* $f$ by

$$f(V_1, \ldots, V_l) = \rho\left([\phi_{d_i}(V_i V_i^\top)]_{i=1}^l\right), \tag{7}$$

where each $\phi_{d_i}$ is shared amongst all subspaces of the same dimension $d_i$, and $l$ is the number of eigenspaces (i.e., number of distinct eigenvalues, which can differ from the number of eigenvectors $k$). As $l$ differs between graphs, we may use zero-padding or a sequence model like a Transformer to parameterize $\rho$. Again, $\phi_{d_i}$ and $\rho$ are generally unrestricted neural networks. To obtain permutation equivariance, we make $\rho$ permutation equivariant and let $\phi_{d_i} = \mathrm{IGN}_{d_i} : \mathbb{R}^{n^2} \to \mathbb{R}^n$ be IGNs from matrices to vectors. For efficiency, we will only use matrices and vectors in the IGNs (that is, no tensors in $\mathbb{R}^{n^p}$ for $p > 2$), i.e., we use 2-IGN. Our resulting *BasisNet* is

$$f(V_1, \ldots, V_l) = \rho\left([\mathrm{IGN}_{d_i}(V_i V_i^\top)]_{i=1}^l\right). \tag{8}$$

**Expressive-BasisNet.** While we restrict SignNet to only use vectors and BasisNet to only use vectors and matrices, higher order tensors are generally required for universally approximating permutation equivariant or invariant functions [Keriven and Peyré, 2019, Maron et al., 2019, Maehara and NT, 2019]. Thus, we will consider a theoretically powerful but computationally impractical variant of our model, in which we replace $\rho$ and $\mathrm{IGN}_{d_i}$ in BasisNet with IGNs of arbitrary tensor order. We call this variant *Expressive-BasisNet*. Universal approximation requires $\Omega(n^n)$ sized intermediate tensors [Ravanbakhsh, 2020]. We study Expressive-BasisNet due to its theoretical interest, and to juxtapose with the computational efficiency and strong expressive power of SignNet and BasisNet.

For a summary of properties and more details about our models, see Appendix B.

In the multiple subspace case, we can prove universality of our models through a general decomposition theorem, which reduces the multiple subspace case to the single subspace case. See Section A for details; we have temporarily moved this Section in the revision due to space constraints, and we will move this Section into the main paper in the camera-ready version.

## 3 Theoretical Power for Graph Representation Learning

Next, we establish that our SignNet and BasisNet can compute useful basis invariant and permutation equivariant functions on Laplacian eigenvectors for graph representation learning, including: spectral

graph convolutions, spectral invariants, and existing graph positional encodings. Expressive-BasisNet can of course compute these functions, as it is universal, but this section shows that the practical invariant architectures SignNet and BasisNet can compute them as well.

## 3.1 SignNets and BasisNets Generalize Spectral Graph Convolution

For node features $X \in \mathbb{R}^{n \times q}$ and an eigendecomposition $V \Lambda V^\top$, a *spectral graph convolution* takes the form $f(V, \Lambda, X) = \sum_{i=1}^{n} \theta_i v_i v_i^\top X = V \operatorname{Diag}(\theta) V^\top X$, for some parameters $\theta_i$, that may optionally be continuous functions $h(\lambda_i) = \theta_i$ of the eigenvalues [Bruna et al., 2014, Defferrard et al., 2016]. This family includes important functions like heat kernels and generalized PageRanks on graphs [Li et al., 2019]. A spectral GNN is defined as multiple layers of spectral graph convolutions and node-wise linear maps, e.g. $V \operatorname{Diag}(\theta_2) V^\top \sigma \left( V \operatorname{Diag}(\theta_1) V^\top X W_1 \right) W_2$ is a two layer spectral GNN. It can be seen (in Appendix H.1) that spectral graph convolutions are permutation equivariant and sign invariant, and if $\theta_i = h(\lambda_i)$ (i.e. the spectral graph convolution is parametric) they are additionally invariant to a change of bases in each eigenspace.

Our SignNet and BasisNet can be viewed as generalizations of spectral graph convolutions, as our networks can universally approximate all spectral graph convolutions of the above form. For instance, SignNet with $\rho(a_1, \ldots, a_k) = \sum_{i=1}^{k} a_k$ and $\phi(v_i, \lambda_i, X) = \frac{1}{2} \theta_i v_i v_i^\top X$ directly yields the spectral graph convolution. This is captured in Theorem 1, which we prove in Appendix H.1. In fact, we may expect SignNet to learn spectral graph convolutions well, according to the principle of algorithmic alignment [Xu et al., 2020] (see Appendix H.1); this is supported by numerical experiments in Appendix J.2, in which our networks outperform baselines in learning spectral graph convolutions.

**Theorem 1.** *SignNet universally approximates all spectral graph convolutions. BasisNet universally approximates all parametric spectral graph convolutions.*

In fact, SignNet and BasisNet are strictly stronger than spectral graph convolutions; there are functions computable by SignNet and BasisNet that cannot be approximated by spectral graph convolutions or spectral GNNs. One way to see this is through graph isomorphism power, as captured in this next result.

**Proposition 3.** *There exist infinitely many pairs of non-isomorphic graphs that SignNet and BasisNet can distinguish, but spectral graph convolutions or spectral GNNs cannot distinguish.*

## 3.2 BasisNets can Compute Spectral Invariants

Many works measure the expressive power of graph neural networks by comparing their power for testing graph isomorphism [Xu et al., 2019, Sato, 2020], or by comparing their ability to compute certain functions on graphs like subgraph counts [Chen et al., 2020, Tahmasebi et al., 2020]. These works often compare GNNs to combinatorial invariants on graphs, especially the $k$-Weisfeiler-Lehman ($k$-WL) tests of graph isomorphism [Morris et al., 2021].

While we may also compare with these combinatorial invariants, as other GNN works that use spectral information have done [Beaini et al., 2021], we argue that it is more natural to analyze our networks in terms of *spectral invariants*, which are computed from the eigenvalues and eigenvectors of graphs. There is a rich literature of spectral invariants from the fields of spectral graph theory and complexity theory [Cvetković et al., 1997]. A spectral invariant must be invariant to permutations and changes of basis in each eigenspace, a characteristic shared by our networks.

The simplest spectral invariant is the multiset of eigenvalues, which we give as input to our networks. Another widely studied, powerful spectral invariant is the collection of graph angles, which are defined as the values $\alpha_{ij} = \|V_i V_i^\top e_j\|_2$, where $V_i \in \mathbb{R}^{n \times d_i}$ is an orthonormal basis for the $i$th adjacency matrix eigenspace, and $e_j$ is the $j$th standard basis vector, which is zero besides a one in the $j$th component. These are easily computed by our networks (Appendix H.3), so our networks inherit the strength of these invariants. We capture these results in the following theorem, which also lists a few properties that graph angles determine [Cvetković, 1991].

**Theorem 2.** *BasisNet universally approximates the graph angles $\alpha_{ij}$. The eigenvalues and graph angles (and thus BasisNet) can determine the number of length 3, 4, or 5 cycles, whether a graph is connected, and the number of length $k$ closed walks from any vertex to itself.*

**Relation to WL and message passing.** In contrast to this result, message passing GNNs are not able to express any of these properties (see [Arvind et al., 2020, Garg et al., 2020] and Appendix H.3). Although spectral invariants are strong, Fürer [2010] shows that the eigenvalues and graph angles—as well as some strictly stronger spectral invariants—are not stronger than the 3-WL test (or, equivalently, the 2-Folklore-WL test). Future work could study the combination of spectral invariants or spectral graph positional encodings with combinatorial algorithms and graph neural networks.

### 3.3 SignNets and BasisNets Generalize Existing Graph Positional Encodings

Many graph positional encodings have been proposed, without any clear criteria on which to choose for a particular task. We prove (in Appendix H.2) that our efficient SignNet and BasisNet can universally approximate many previously used graph positional encodings, because we unify these positional encodings by expressing them as either a spectral graph convolution matrix or the diagonal of a spectral graph convolution matrix.

**Proposition 4.** *SignNet and BasisNet universally approximate node positional encodings based on heat kernels [Feldman et al., 2022] and random walks [Dwivedi et al., 2022]. BasisNet universally approximates diffusion and $p$-step random walk relative positional encodings [Mialon et al., 2021], and generalized PageRank and landing probability distance encodings [Li et al., 2020].*

We note that diagonals of spectral convolutions are used as feature descriptors in the shape analysis literature, such as the heat kernel signature [Sun et al., 2009] and wave kernel signature [Aubry et al., 2011]. In the language of recent works in graph machine learning, these are node positional encodings computed from a discrete Laplacian of a triangle mesh. This connection appears to be unnoticed in recent works on graph positional encodings.

## 4 Experiments

We demonstrate the strength of our networks in various experiments. Appendix B shows simple pseudo-code and a diagram detailing the use of SignNet as a node positional encoding.

### 4.1 Graph Regression

Table 1: Results on the ZINC dataset with a 500k parameter budget. All models use edge features. Numbers are the mean and standard deviation over 4 runs, each with different seeds.

| Base model | Positional encoding | $k$ | #param | Test MAE ($\downarrow$) |
|---|---|---|---|---|
| GatedGCN | No PE | N/A | 492k | $0.252_{\pm 0.007}$ |
| | LapPE (flip) | 8 | 492k | $0.198_{\pm 0.011}$ |
| | LapPE (abs.) | 8 | 492k | $0.204_{\pm 0.009}$ |
| | LapPE (can.) | 8 | 505k | $0.298_{\pm 0.019}$ |
| | SignNet ($\phi(v)$ only) | 8 | 495k | $0.148_{\pm 0.007}$ |
| | SignNet | 8 | 495k | $0.121_{\pm 0.005}$ |
| | SignNet | All | 491k | $\mathbf{0.100_{\pm 0.007}}$ |
| Sparse Transformer | No PE | N/A | 473k | $0.283_{\pm 0.030}$ |
| | LapPE (flip) | 16 | 487k | $0.223_{\pm 0.007}$ |
| | SignNet | 16 | 479k | $0.115_{\pm 0.008}$ |
| | SignNet | All | 486k | $\mathbf{0.102_{\pm 0.005}}$ |
| GINE | No PE | N/A | 470k | $0.170_{\pm 0.002}$ |
| | LapPE (flip) | 16 | 470k | $0.178_{\pm 0.004}$ |
| | SignNet | 16 | 470k | $0.147_{\pm 0.005}$ |
| | SignNet | All | 417k | $\mathbf{0.102_{\pm 0.002}}$ |
| PNA | No PE | N/A | 474k | $0.133_{\pm 0.011}$ |
| | LapPE (flip) | 8 | 474k | $0.132_{\pm 0.010}$ |
| | SignNet | 8 | 476k | $0.105_{\pm 0.007}$ |
| | SignNet | All | 487k | $\mathbf{0.084_{\pm 0.006}}$ |

We study the effectiveness of SignNet for learning positional encodings (PEs) from the eigenvectors of the graph Laplacian on the ZINC dataset of molecule graphs [Irwin et al., 2012] (using the

Table 2: Comparison with SOTA methods on graph-level regression tasks. † denotes domain-specific model. Numbers are test MAE, so lower is better. Best models within a standard deviation are bolded.

| | ZINC (10K) ↓ | ZINC-full ↓ | Alchemy (10k) ↓ |
|---|---|---|---|
| HIMP † [Fey et al., 2020] | $.151_{\pm.006}$ | $.036_{\pm.002}$ | — |
| CIN-small † [Bodnar et al., 2021] | $.094_{\pm.004}$ | $.044_{\pm.003}$ | — |
| CIN † [Bodnar et al., 2021] | $\mathbf{.079}_{\pm.006}$ | $\mathbf{.022}_{\pm.002}$ | — |
| GIN [Xu et al., 2019] | $.170_{\pm.002}$ | $.088_{\pm.002}$ | $.180_{\pm.006}$ |
| $\delta$-2-GNN [Morris et al., 2020b] | $.374_{\pm.022}$ | $.042_{\pm.003}$ | $.118_{\pm.001}$ |
| $\delta$-2-LGNN [Morris et al., 2020b] | $.306_{\pm.044}$ | $.045_{\pm.006}$ | $.122_{\pm.003}$ |
| SpeqNet [Morris et al., 2022] | — | — | $\mathbf{.115}_{\pm.001}$ |
| GNN-IR [Dupty and Lee, 2022] | $.137_{\pm.010}$ | — | $.119_{\pm.002}$ |
| PF-GNN [Dupty et al., 2021] | $.122_{\pm.01}$ | — | $\mathbf{.111}_{\pm.01}$ |
| Recon-GNN [Cotta et al., 2021] | $.170_{\pm.006}$ | — | $.125_{\pm.001}$ |
| SignNet (ours) | $\mathbf{.084}_{\pm.006}$ | $\mathbf{.024}_{\pm.003}$ | $\mathbf{.113}_{\pm.002}$ |

subset of 12,000 graphs from Dwivedi et al. [2020]). We primarily consider three settings: 1) No positional encoding, 2) Laplacian PE (LapPE)—the $k$ eigenvectors of the graph Laplacian with smallest eigenvalues are concatenated with existing node features, 3) SignNet positional features— passing the eigenvectors through a SignNet and concatenating the output with node features. We parameterize SignNet by taking $\phi$ to be a GIN [Xu et al., 2019] and $\rho$ to be an MLP. We sum over $\phi$ outputs before the MLP when handling variable numbers of eigenvectors, so then the SignNet is of the form $\text{MLP}\left(\sum_{i=1}^{l}\phi(v_i) + \phi(-v_i)\right)$ (see Appendix K.2 for further details). We consider four different base models that process the graph data and positional encodings: GatedGCN [Bresson and Laurent, 2017], a Transformer with sparse attention only over neighbours [Kreuzer et al., 2021], PNA [Corso et al., 2020], and GIN [Xu et al., 2019] with edge features (i.e. GINE) [Hu et al., 2020b]. The total number of parameters of the SignNet and the base model is kept within a 500k budget.

Table 1 shows the results. For all 4 base models, the PE learned with SignNet yields the best test MAE (mean absolute error) — lower MAE is better. Notably, this includes the cases of PNA and GINE, for which Laplacian PE with simple random sign flipping was unable to improve performance over using no PE at all. Our best performing model is PNA base combined with SignNet, which achieves $0.084$ test MAE. Besides SignNet, we consider two non-learned approaches to resolving eigenvector sign ambiguity—canonicalization and taking element-wise absolute values (see Appendix K.2 for details). Results with GatedGCN show that these alternatives are not more effective than random sign flipping for learning positional encodings. We also consider an ablation of our SignNet architecture where we remove the sign invariance, using simply $\text{MLP}([\phi(v_i)]_{i=1}^{k})$. Although the resulting architecture is no longer sign invariant, $\phi$ still processes eigenvectors independently, meaning that only two invariances ($\pm 1$) need be learned, significantly fewer than the $2^k$ total sign flip configurations. Accordingly, this non-sign invariant learned positional encoding achieves a test MAE of $0.148$, improving over the Laplacian PE ($0.198$) but falling short of the fully sign invariant SignNet ($0.121$). In all cases, using all available eigenvectors in SignNet significantly improves performance over using a fixed number of eigenvectors. In Appendix J.1, we also show that SignNet improves performance when no edge features are included in the data.

These significant performance improvements from SignNet come with only a slightly higher computational cost. For example, GatedGCN with no PE takes about 8.2 seconds per training iteration on ZINC, while GatedGCN with 8 eigenvectors and SignNet takes about 10.6 seconds; this is only a 29% increase in time, for a reduction of test MAE by over 50%. Also, eigenvector computation time is neglible, we need only precompute and save the eigenvectors once, and it only takes 15 seconds to do this for the 12,000 graphs of ZINC.

**Comparison with SOTA.** In Table 2, we compare SignNet with state-of-the-art methods on graph-level molecular regression tasks on ZINC (10,000 training graphs), ZINC-full (about 250,000 graphs), and Alchemy [Chen et al., 2019a] (10,000 training graphs). We compare against both methods that use domain-specific knowledge about molecules, and domain-agnostic GNNs of various architectures. We see that SignNet outperforms all domain-agnostic methods on ZINC and ZINC-full, and is within a standard deviation of the best domain-specific method. Our mean score is the second best on

Table 3: Test results for texture reconstruction experiment on cat and human models, following the experimental setting of [Koestler et al., 2022]. We use 1023 eigenvectors of the cotangent Laplacian.

| Method | Params | Cat | | | Human | | |
| --- | --- | --- | --- | --- | --- | --- | --- |
| | | PSNR ↑ | DSSIM ↓ | LPIPS ↓ | PSNR ↑ | DSSIM ↓ | LPIPS ↓ |
| Intrinsic NF | 329k | 34.25 | .099 | .189 | 32.29 | **.119** | .330 |
| Absolute value | 329k | 34.67 | .106 | .252 | **32.42** | .132 | .363 |
| Sign flip | 329k | 23.15 | 1.28 | 2.35 | 21.52 | 1.05 | 2.71 |
| SignNet | 324k | **34.91** | **.090** | **.147** | **32.43** | .125 | **.316** |

Alchemy, and is within a standard deviation of the best. We perform much better on ZINC (.084) than other state-of-the-art positional encoding methods, like GNN-LSPE (.090) [Dwivedi et al., 2022], SAN (.139) [Kreuzer et al., 2021], and Graphormer (.122) [Ying et al., 2021].

## 4.2 Counting Substructures and Regressing Graph Properties

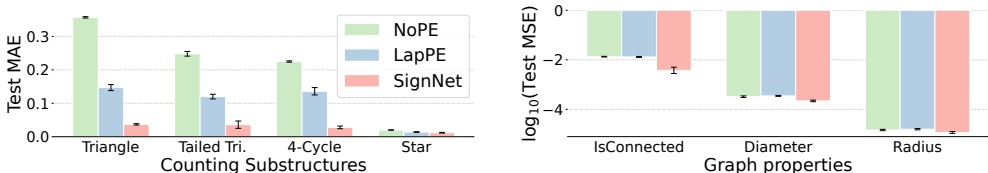

Figure 3: Counting substructures and regressing graph properties (lower is better). With Laplacian PEs, SignNet improves performance, while sign flip data augmentation (LapPE) is less consistent. Mean and standard deviations are reported on 3 runs. All runs use the same 4-layer GIN base model.

Substructure counts (e.g. of cycles) and global graph properties (e.g. connectedness, diameter, radius) are important graph features that are known to be informative for problems in bio- and chemo-informatics [Chen et al., 2020, Corso et al., 2020]. Following the setting of Zhao et al. [2022], we show that SignNet with Laplacian positional encodings boosts the ability of simple GNNs to count substructures and regress graph properties. We take a 4-layer GIN as the base model for all settings, and for SignNet we use GIN as $\phi$ and a Transformer as $\rho$ to handle variable numbers of eigenvectors (see Appendix K.4 for details). As shown in Figure 3, Laplacian PEs with sign-flip data augmentation improve performance for counting substructures but not for regressing graph properties, while Laplacian PEs processed by SignNet significantly boost performance on all tasks.

## 4.3 Neural Fields on Manifolds

Discrete approximations to the Laplace-Beltrami operator on manifolds have proven useful for processing data on surfaces, such as triangle meshes [Lévy, 2006]. Recently, Koestler et al. [2022] propose intrinsic neural fields, which use eigenfunctions of the Laplace-Beltrami operator as positional encodings for learning neural fields on manifolds. For generalized eigenfunctions $v_1, \ldots, v_k$, at a point $p$ on the surface, they parameterize functions $f(p) = \text{MLP}(v_1(p), \ldots, v_k(p))$. As these eigenfunctions have sign ambiguity, we use our SignNet to parameterize $f(p) = \text{MLP}( \rho( [\phi(v_i(p)) + \phi(-v_i(p))]_{i=1,\ldots,k} ) )$, with $\rho$ and $\phi$ being MLPs.

Table 3 shows our results for texture reconstruction experiments on all models from Koestler et al. [2022]. The total number of parameters in our SignNet-based model is kept below that of the original model. We see that the SignNet architecture improves over the original Intrinsic NF model and over other baselines — especially in the LPIPS (Learned Perceptual Image Patch Similarity) metric, which has been shown to be a typically better perceptual metric than PSNR or DSSIM [Zhang et al., 2018a]. While we have not yet tested this, we believe that SignNet would allow even better improvements when learning over eigenfunctions of different models, as it could improve transfer and generalization. See Appendix D.1 for visualizations and Appendix K.5 for more details.

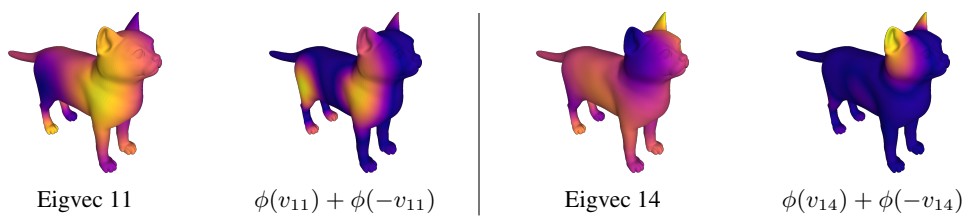

| Eigvec 11 | $\phi(v_{11}) + \phi(-v_{11})$ | Eigvec 14 | $\phi(v_{14}) + \phi(-v_{14})$ |

Figure 4: Cotangent Laplacian eigenvectors of the cat model and first principal component of $\phi(v) + \phi(-v)$ from our trained SignNet.

## 4.4 Visualization of Learned Positional Encodings

To better understand SignNet, we plot the first principal component of $\phi(v) + \phi(-v)$ for two eigenvectors on the cat model in Figure 4. We see that SignNet encodes bilateral symmetry and structural information on the cat model. See Appendix D for plots of more eigenvectors and further details.

## 5 Related Work

In this section, we review selected related work. A more thorough review is deferred to Appendix E.

**Laplacian eigenvectors in GNNs.** Various recently proposed methods in graph deep learning have directly used Laplacian eigenvectors as node positional encodings that are input to a neural network that is, e.g., a message passing GNN [Dwivedi et al., 2020, 2022], or some variant of a Transformer that is adapted to graphs [Dwivedi and Bresson, 2021, Kreuzer et al., 2021, Mialon et al., 2021, Dwivedi et al., 2022]. None of these methods address basis invariance, and they only partially address sign invariance for node positional encodings by randomly flipping eigenvector signs during training.

**Graph positional encodings.** Other recent methods use positional encodings besides Laplacian eigenvectors. These include positional encodings based on random walks [Dwivedi et al., 2022, Mialon et al., 2021, Li et al., 2020], diffusion kernels on graphs [Mialon et al., 2021, Feldman et al., 2022], shortest paths [Ying et al., 2021, Li et al., 2020], and unsupervised node embedding methods [Wang et al., 2022]. In particular, Wang et al. [2022] use Laplacian eigenvectors for relative positional encodings in an invariant way, but they focus on robustness, so they have stricter invariances that significantly reduce expressivity (see Appendix E.2 for more details). These previously used positional encodings are mostly ad-hoc, less general since they can be provably expressed by SignNet and BasisNet (see Section 3.3), and/or are expensive to compute (e.g., all pairs shortest paths).

## 6 Conclusion and Discussion

SignNet and BasisNet are novel architectures for processing eigenvectors that are invariant to sign flips and choices of eigenspace bases, respectively. Both architectures are provably universal: they can represent any continuous function with the corresponding invariances. When used with Laplacian eigenvectors as inputs they can provably approximate spectral graph convolutions, spectral invariants, graph properties such as subgraph counts, and a number of other graph positional encodings. These theoretical results are supported by experiments showing that SignNet and BasisNet are highly expressive in practice, and learn effective graph positional encodings that improve the performance of message passing graph neural networks. Initial explorations show that SignNet and BasisNet can be useful beyond graph representation learning, as eigenvectors are ubiquitous.

While we conduct experiments on graph machine learning tasks and a particular task on triangle meshes, SignNet and BasisNet should also be applicable to processing eigenvectors in other settings, such as recommender systems and tasks in shape analysis. We show significant empirical benefit in the tasks that we consider, but we expect less benefit in cases where node features are sufficient to do well on the task, or if the task does not require much sophisticated graph structure information to solve. Moreover, while we primarily consider eigenspaces in this work, sign invariance and basis invariance applies to any model that processes subspaces of a vector space; future work may explore our models on general subspaces.

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

# A  Universality for Multiple Spaces

While the networks introduced in the Section 2.2 possess the desired invariances, it is not immediately obvious whether they are powerful enough to express *all* functions with these invariances. The universality of our architectures follows as a corollary of the following general decomposition result, which may enable construction of universal architectures for other invariances as well.

**Theorem 3** (Decomposition Theorem). *Let $\mathcal{X}_1, \ldots, \mathcal{X}_k$ be topological spaces, and let $G_i$ be a group acting on $\mathcal{X}_i$ for each $i$. We assume mild topological conditions on $\mathcal{X}_i$ and $G_i$ hold. For any continuous $f : \mathcal{X} = \mathcal{X}_1 \times \ldots \times \mathcal{X}_k \to \mathbb{R}^s$ that is invariant to the action of $G = G_1 \times \ldots \times G_k$, there exists continuous $\phi_i$ and a continuous $\rho : \mathcal{Z} \subseteq \mathbb{R}^a \to \mathbb{R}^s$ such that*

$$f(v_1, \ldots, v_k) = \rho(\phi_1(v_1), \ldots, \phi_k(v_k)). \tag{9}$$

*Furthermore: (1) each $\phi_i$ can be taken to be invariant to $G_i$, (2) the domain $\mathcal{Z}$ of $\rho$ is compact if each $\mathcal{X}_i$ is compact, (3) if $\mathcal{X}_i = \mathcal{X}_j$ and $G_i = G_j$, then $\phi_i$ can be taken to be equal to $\phi_j$.*

This result says that when a product of groups $G$ acts on a product of spaces $\mathcal{X}$, for invariance to the product group $G$ it suffices to individually process each smaller group $G_i$ on $\mathcal{X}_i$ and then aggregate the results. Along with the proof of Theorem 3, the mild topological assumptions are explained in Appendix G.1. The assumptions hold for sign invariance and basis invariance. By applying this theorem, we can prove universality of our networks:

**Corollary 1.** *Unconstrained-SignNet can represent any sign invariant function and Unconstrained-BasisNet can represent any basis invariant function. Expressive-BasisNet is a universal approximator of functions that are both basis invariant and permutation equivariant.*

This result shows that Unconstrained-SignNet, Unconstrained-BasisNet, and Expressive-BasisNet take the correct functional form for their respective invariances (proofs in Appendix G.2). Note that Expressive-BasisNet approximates all sign invariant functions as a special case, by treating all inputs as one dimensional eigenspaces. Accompanying the decomposition result, we show a corresponding universal approximation result (proof in Appendix G.3). Similarly to Theorem 3, the problem of approximating $G = G_1 \times \ldots \times G_k$ invariant functions is reduced to approximating several $G_i$-invariant functions.

# B  More Details on SignNet and BasisNet

Table 4: Properties of our architectures: Unconstrained-SignNet, SignNet, Unconstrained-BasisNet, and Expressive-BasisNet. The properties are: permutation equivariance, universality (for the proper class of continuous invariant functions), and computational tractability.

|                    | Unconstr.-SignNet | SignNet | Unconstr.-BasisNet | BasisNet | Expr.-BasisNet |
|--------------------|:-----------------:|:-------:|:------------------:|:--------:|:--------------:|
| Permutation equiv. | ✗                 | ✓       | ✗                  | ✓        | ✓              |
| Universal          | ✓                 | ✗       | ✓                  | ✗        | ✓              |
| Tractable          | ✓                 | ✓       | ✓                  | ✓        | ✗              |

In Figure 2, we show a diagram that describes how SignNet is used as a node positional encoding for a graph machine learning task. In Table 4, we compare and contrast properties of the neural architectures that we introduce. In Figure 5, we give pseudo-code of SignNet for learning node positional encodings with a GNN prediction model.

## B.1  Generalization Beyond Symmetric Matrices

In the main paper, we assume that the eigenspaces come from a symmetric matrix. This holds for many cases of practical interest, as e.g. the Laplacian matrix of an undirected graph is symmetric. However, we may also want to process directed graphs, or other data that have associated nonsymmetric matrices. Our SignNet and BasisNet generalize in a straightforward way to handle nonsymmetric diagonalizable

PyTorch-like pseudo-code for SignNet

```python
class SignNetGNN(nn.Module):

    def __init__(self, d, k, D1, D2, out_dim):
        self.phi = GIN(1, D1) # in dim=1, out dim=D1
        self.rho = MLP(k*D1, D2)
        self.base_model = GNN(d+D2, out_dim)

    def forward(self, g, x, eigvecs):
        # g contains graph information
        # x shape: n x d
        # eigvecs shape: n x k

        n, k = eigvecs.shape
        eigvecs = eigvecs.reshape(n, k, 1)
        pe = self.phi(g, eigvecs) + self.phi(g, -eigvecs)
        pe = pe.reshape(n, -1) # n x k x D1 -> n x k*D1
        pe = self.rho(pe)

        return self.base_model(g, x, pe)
```

Figure 5: PyTorch-like pseudo-code for using SignNet with a GNN prediction model, where $\phi = $ GIN and $\rho = $ MLP as in the ZINC molecular graph regression experiments. Reshaping eigenvectors from $n \times k$ to $n \times k \times 1$ allows $\phi$ to process each eigenvector (and its negation) independently in PyTorch-like deep learning libraries.

matrices, as we detail here. Let $A \in \mathbb{R}^{n \times n}$ be a matrix with a diagonalization $A = V \Lambda V^{-1}$, where $\Lambda = \text{Diag}(\lambda_1, \ldots, \lambda_n)$ contains the eigenvalues $\lambda_i$, and the columns of $V = [v_1 \ \ldots \ v_n]$ are eigenvectors. Suppose we want to learn a function on the eigenvectors $v_1, \ldots, v_k$. Unlike in the symmetric matrix case, the eigenvectors are not necessarily orthonormal, and both the eigenvalues and eigenvectors can be complex.

**Real eigenvectors.** First, we assume the eigenvectors $v_i$ are all real vectors in $\mathbb{R}^n$. We can take the eigenvectors to be real if $A$ is symmetric, or if $A$ has real eigenvalues (see Horn and Johnson [2012] Theorem 1.3.29). Also, suppose that we choose the real numbers $\mathbb{R}$ as our base field for the vector space in which eigenvectors lie. Note that for any scaling factor $c \in \mathbb{R} \setminus \{0\}$ and eigenvector $v$, we have that $cv$ is an eigenvector of the same eigenvalue. If the eigenvalues are distinct, then the eigenvectors of the form $cv$ are the only other eigenvectors in the same eigenspace as $v$. Thus, we want a function to be invariant to scalings:

$$f(v_1, \ldots, v_k) = f(c_1 v_1, \ldots, c_k v_k) \qquad c_i \in \mathbb{R} \setminus \{0\}. \tag{10}$$

This can be handled by SignNet, by giving unit normalized vector inputs:

$$f(v_1, \ldots, v_k) = \rho \left( [\phi(v_i/\|v_i\|) + \phi(-v_i/\|v_i\|)]_{i=1,\ldots,k} \right). \tag{11}$$

Now, say have bases of eigenspaces $V_1, \ldots, V_l$ with dimensions $d_1, \ldots, d_l$. For a basis $V_i$, we have that any other basis of the same space can be obtained as $V_i W$ for some $W \in \text{GL}_{\mathbb{R}}(d_i)$, the set of real invertible matrices in $\mathbb{R}^{d_i \times d_i}$. Indeed, the orthonormal projector for the space spanned by the columns of $V_i$ is given by $V_i(V_i^\top V_i)^{-1} V_i^\top$. Thus, if $Z \in \mathbb{R}^{n \times d_i}$ is another basis for the column space of $V_i$, we have that $V_i(V_i^\top V_i)^{-1} V_i^\top = Z(Z^\top Z)^{-1} Z^\top$, so

$$V_i(V_i^\top V_i)^{-1} V_i^\top Z = Z(Z^\top Z)^{-1} Z^\top Z = Z, \tag{12}$$

so let $W = (V_i^\top V_i)^{-1} V_i^\top Z \in \mathbb{R}^{d_i \times d_i}$. Note that $W$ is invertible, because it has inverse $(Z^\top Z)^{-1} Z^\top V_i$, so indeed $V_i W = Z$ for $W \in \text{GL}_{\mathbb{R}}(d_i)$. Thus, basis invariance in this case is of the form

$$f(V_1 \ldots, V_l) = f(V_1 W_1, \ldots, V_l W_l) \qquad W_i \in \text{GL}_{\mathbb{R}}(d_i). \tag{13}$$

Note that the distinct eigenvalue invariance is a special case of this invariance, as $\text{G}_{\mathbb{R}}(1) = \mathbb{R} \setminus \{0\}$. We can again achieve this basis invariance by using a BasisNet, where the inputs to the $\phi_{d_i}$ are orthogonal projectors of the corresponding eigenspace:

$$f(V_1, \ldots, V_l) = \rho \left( [\phi_{d_i}(V_i(V_i^\top V_i)^{-1} V_i^\top)]_{i=1,\ldots,l} \right). \tag{14}$$

Recall that if $V_i$ is an orthonormal basis, then the orthogonal projector is just $V_i V_i^\top$, so this is a direct generalization of BasisNet in the symmetric case.

**Complex eigenvectors.** More generally, suppose $V \in \mathbb{C}^{n \times n}$ are complex eigenvectors, and we take the base field of the vector space to be $\mathbb{C}$. The above arguments generalize to the complex case; in the case of distinct eigenvalues, we want

$$f(v_1, \ldots, v_k) = f(c_1 v_1, \ldots, c_k v_k) \qquad c_i \in \mathbb{C} \setminus \{0\}. \tag{15}$$

However, this symmetry can not be as easily reduced to a unit normalization and a discrete sign invariance, as it can be in the real case. Nonetheless, the basis invariant architecture directly generalizes, so we can handle the case of distinct eigenvalues by a more general basis invariant architecture as well. The basis invariance is

$$f(V_1, \ldots, V_l) = f(V_1 W_1, \ldots, V_l W_l) \qquad W_i \in \mathrm{GL}_{\mathbb{C}}(d_i). \tag{16}$$

The orthogonal projector of the image of $V_i$ is $V_i (V_i^* V_i)^{-1} V_i^*$, where there are now conjugate transposes replacing the transposes. Thus, BasisNet takes the form:

$$f(V_1, \ldots, V_l) = \rho \left( \left[ \phi_{d_i} (V_i (V_i^* V_i)^{-1} V_i^*) \right]_{i=1,\ldots,l} \right). \tag{17}$$

## B.2 Broader Impacts

We believe that our models and future sign invariant or basis invariant networks could be useful in a wide variety of applications. As eigenvectors arise in many domains, it is difficult to predict the uses of these models. We test on several molecular property prediction tasks, which have the potential for much positive impact, such as in drug discovery [Stokes et al., 2020]. However, recent work has found that the same models that we use for finding beneficial drugs can also be used to design biochemical weapons [Urbina et al., 2022]. Another major application of graph machine learning is in social network analysis, where positive (e.g. malicious node detection [Pandit et al., 2007]) and negative (e.g. deanonymization [Narayanan and Shmatikov, 2009]) uses of machine learning are possible. Even if there is no negative intent, bias in learned models can differentially impact particular subgroups of people. Thus, academia, industry, and policy makers must be aware of such potential negative uses, and work towards reducing the likelihood of them.

## C    More on Eigenvalue Multiplicities

In this section, we study the properties of eigenvalues and eigenvectors computed by numerical algorithms on real-world data.

### C.1    Sign and Basis Ambiguities in Numerical Eigensolvers

When processing real-world data, we use eigenvectors that are computed by numerical algorithms. These algorithms return specific eigenvectors for each eigenspace, so there is some choice of sign or basis of each eigenspace. The general symmetric matrix eigensolvers `numpy.linalg.eigh` and `scipy.linalg.eigh` both call LAPACK routines. They both proceed as follows: for a symmetric matrix $A$, they first decompose it as $A = QTQ^\top$ for orthogonal $Q$ and tridiagonal $T$, then they compute the eigendecomposition of $T = W \Lambda W^\top$, so the eigendecomposition of $A$ is $A = (QW) \Lambda (W^\top Q^\top)$. There are multiple ambiguities here: for diagonal sign matrices $S = \mathrm{Diag}(s_1, \ldots, s_n)$ and $S' = \mathrm{Diag}(s'_1, \ldots, s'_n)$, where $s_i, s'_i \in \{-1, 1\}$, we have that $A = QS(STS)SQ^\top$ is also a valid tridiagonalization, as $QS$ is still orthogonal, $SS = I$, and $STS$ is still tridiagonal. Also, $T = (WS') \Lambda (S'W^\top)$ is a valid eigendecomposition of $T$, as $WS'$ is still orthogonal.

In practice, we find that the general symmetric matrix eigensolvers `numpy.linalg.eigh` and `scipy.linalg.eigh` differ between frameworks but are consistent with the same framework. More specifically, for a symmetric matrix $A$, we find that the eigenvectors computed with the default settings in numpy tend to differ by a choice of sign or basis from those that are computed with the default settings in scipy. On the other hand, the called LAPACK routines are deterministic, so the eigenvectors returned by numpy are the same in each call, and the eigenvectors returned by scipy are likewise the same in each call.

Table 5: Eigenspace statistics for datasets of multiple graphs. From left to right, the columns are: dataset name, number of graphs, range of number of nodes per graph, largest multiplicity, and percent of graphs with an eigenspace of dimension $> 1$.

| Dataset | Graphs | # Nodes | Max. Mult | % Graphs mult. $> 1$ |
|---|---|---|---|---|
| ZINC | 12,000 | 9-37 | 9 | 64.1 |
| ZINC-full | 249,456 | 6-38 | 10 | 63.8 |
| ogbg-molhiv | 41,127 | 2 - 222 | 42 | 68.0 |
| IMDB-M | 1,500 | 7 - 89 | 37 | 99.9 |
| COLLAB | 5,000 | 32 - 492 | 238 | 99.1 |
| PROTEINS | 1,113 | 4 - 620 | 20 | 77.3 |
| COIL-DEL | 3,900 | 3 - 77 | 4 | 4.00 |

Eigensolvers for sparse symmetric matrices like `scipy.linalg.eigsh` are required for large scale problems. This function calls ARPACK, which uses an iterative method that starts with a randomly sampled initial vector. Due to this stochasticity, the sign and basis of eigenvectors returned differs between each call.

Bro et al. [2008] develops a data-dependent method to choose signs for each singular vector of a singular value decomposition. Still, in the worst case the signs chosen will be arbitrary, and they do not handle basis ambiguities in higher dimensional eigenspaces. Other works have made choices of sign, such as by picking the sign so that the eigenvector's entries are in the largest lexicographic order [Tam and Dunson, 2022]. This choice of sign may work poorly for learning on graphs, as it is sensitive to permutations on nodes. For some graph regression experiments in Section 4.1, we try a choice of sign that is permutation invariant, but we find it to work poorly.

## C.2   Higher Dimensional Eigenspaces in Real Graphs

Here, we investigate the normalized Laplacian eigenspace statistics of real-world graph data. For any graph that has distinct Laplacian eigenvalues, only sign invariance is required in processing eigenvectors. However, we find that graph data tends to have higher multiplicity eigenvalues, so basis invariance would be required for learning symmetry-respecting functions on eigenvectors.

Indeed, we show statistics for multi-graph datasets in Table 5 and for single-graph datasets with more nodes per graph in Table 6. For multi-graph datasets, we consider :

- Molecule graphs: ZINC [Irwin et al., 2012, Dwivedi et al., 2020], ogbg-molhiv [Wu et al., 2018, Hu et al., 2020a]

- Social networks: IMDB-M, COLLAB [Yanardag and Vishwanathan, 2015, Morris et al., 2020a],

- Bioinformatics graphs: PROTEINS [Morris et al., 2020a]

- Computer vision graphs: COIL-DEL [Riesen and Bunke, 2008, Morris et al., 2020a].

For single-graph datasets, we consider:

- The $32 \times 32$ image grid as in Section J.2

- Citation networks: Cora, Citeseer [Sen et al., 2008]

- Co-purchasing graphs with Amazon Photo [McAuley et al., 2015, Shchur et al., 2018].

We see that these datasets all contain higher multiplicity eigenspaces, so sign invariance is insufficient for fully respecting symmetries. The majority of graphs in each multi-graph dataset besides COIL-DEL contain higher multiplicity eigenspaces. Also, the dimension of these eigenspaces can be quite large compared to the size of the graphs in the dataset. The single-graph datasets have a large proportion of their eigenvectors belonging to higher dimensional eigenspaces. Thus, basis invariance may play a large role in processing spectral information from these graph datasets.

Table 6: Eigenspace statistics for single graphs. From left to right, the columns are: dataset name, number of nodes, distinct eigenvalues (i.e. distinct eigenspaces), number of unique multiplicities, largest multiplicity, and percent of eigenvectors belonging to an eigenspace of dimension $> 1$.

| Dataset | Nodes | Distinct $\lambda$ | # Mult. | Max Mult. | % Vecs mult. $> 1$ |
|---|---|---|---|---|---|
| $32 \times 32$ image | 1,024 | 513 | 3 | 32 | 96.9 |
| Cora | 2,708 | 2,187 | 11 | 300 | 19.7 |
| Citeseer | 3,327 | 1,861 | 12 | 491 | 44.8 |
| Amazon Photo | 7,650 | 7,416 | 8 | 136 | 3.71 |

## C.3   Relationship to Graph Automorphisms

Higher multiplicity eigenspaces are related to automorphism symmetries in graphs. For an adjacency matrix $A$, the permutation matrix $P$ is an automorphism of the graph associated to $A$ if $PAP^\top = A$. If $P$ is an automorphism, then for any eigenvector $v$ of $A$ with eigenvalue $\lambda$, we have

$$APv = PAP^\top Pv = PAv = P\lambda v = \lambda Pv, \tag{18}$$

so $Pv$ is an eigenvector of $A$ with the same eigenvalue $\lambda$. If $Pv$ and $v$ are linearly independent, then $\lambda$ has a higher dimensional eigenspace. Thus, under certain additional conditions, automorphism symmetries of graphs lead to repeated eigenvalues [Sachs and Stiebitz, 1983, Teranishi, 2009].

## C.4   Multiplicities in Random Graphs

It is known that almost all random graphs under the Erdős-Renyi model have no repeated eigenvalues in the infinite number of nodes limit [Tao and Vu, 2017]. Likewise, almost all random graphs under the Erdős-Renyi model are asymmetric in the sense of having no nontrivial automorphism symmetries [Erdos and Rényi, 1963]. These results contrast sharply with the high eigenvalue multiplicities that we see in real-world data in Section C.2. Likewise, many types of real-world graph data have been found to possess nontrivial automorphism symmetries [Ball and Geyer-Schulz, 2018]. This demonstrates a potential downside of using random graph models to study real-world data: the eigenspace dimensions and automorphism symmetries of random graphs may not agree with those of real-world data.

# D Visualization of SignNet output

## D.1 Cat Model Visualization

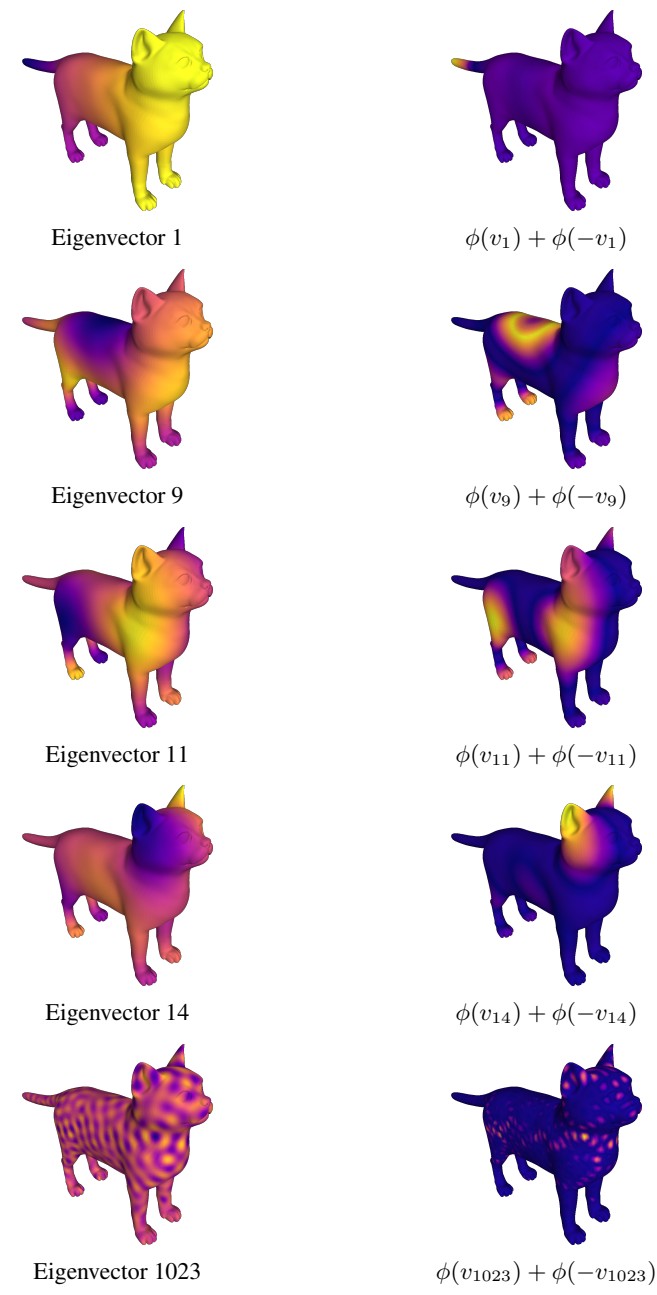

Figure 6: (Left) Cotangent Laplacian eigenvectors of the cat model. (Right) First principal component of $\phi(v) + \phi(-v)$ from our trained SignNet.

In Figure 6, we plot the eigenvectors of the cotangent Laplacian on a cat model, as well as the first principal component of the corresponding learned $\phi(v) + \phi(-v)$ from our SignNet model that was trained on the texture reconstruction task. Interestingly, this portion of our SignNet encodes bilateral symmetry; for instance, while some eigenvectors differ between left feet and right feet, this portion of our SignNet gives similar values for the left and right feet. This is useful for the texture reconstruction task, as the texture regression target has bilateral symmetry.

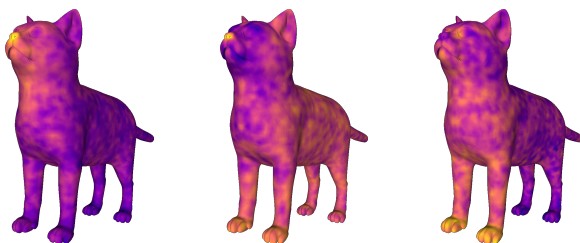

Figure 7: First three principal components of the full SignNet output on the cat model.

We also show principal components of outputs for the full SignNet model in Figure 7. This is not as interpretable, as the outputs are high frequency and appear to be close to the texture that is the regression target. If instead we trained the network on a task involving eigenvectors of multiple models, then we may expect the SignNet to learn more structurally interpretable mappings (as in the case of the molecule tasks).

### D.2 Molecule visualization

To better understand SignNet, in Figure 9 we visualize the learned positional encodings of a SignNet with $\phi = $ GIN, $\rho = $ MLP (with a summation to handle variable eigenvector numbers) trained on ZINC as in Section 4.1. SignNet learns interesting structural information such as min-cuts (PC 3) and appendage atoms (PC 2) that qualitatively differ from any single eigenvector of the graph.

For this visualization we use a SignNet trained with a GatedGCN base model on ZINC, as in Section 4.1. This SignNet uses GIN as $\phi$ and $\rho$ as an MLP (with a sum before it to handle variable numbers of eigenvectors), and takes in all eigenvectors of each graph. See Figure 8 for all of the eigenvectors of fluorescein.

## E  More Related Work

### E.1  Graph Positional Encodings

Various graph positional encodings have been proposed, which have been motivated for increasing expressive power or practical performance of graph neural networks, and for generalizing Transformers to graphs. Positional encodings are related to so-called position-aware network embeddings [Chami et al., 2020], which capture distances between nodes in graphs. These include network embedding methods like Deepwalk [Perozzi et al., 2014] and node2vec [Grover and Leskovec, 2016], which have been recently integrated into GNNs that respect their invariances by Wang et al. [2022]. Further, Li et al. [2020] studies the theoretical and practical benefits of incorporating distance features into graph neural networks. Dwivedi et al. [2022] proposes a method to inject learnable positional encodings into each layer of a graph neural network, and uses a simple random walk based node positional encoding. You et al. [2021] proposes a node positional encoding $\mathrm{diag}(A^k)$, which captures the number of closed walks from a node to itself. Dwivedi et al. [2020] propose to use Laplacian eigenvectors as positional encodings in graph neural networks, with sign ambiguities alleviated by sign flipping data augmentation. Srinivasan and Ribeiro [2019] theoretically analyze node positional embeddings and structural representations in graphs, and show that most-expressive structural representations contain the information of any node positional embedding.

While positional encodings in sequences as used for Transformers [Vaswani et al., 2017] are able to leverage the canonical order in sequences, there is no such useful canonical order for nodes in a graph, due in part to permutation symmetries. Thus, different permutation equivariant positional encodings have been proposed to help generalize Transformers to graphs. Dwivedi and Bresson [2021] directly add in linearly projected Laplacian eigenvectors to node features before processing these features with a graph Transformer. Kreuzer et al. [2021] propose an architecture that uses attention over Laplacian eigenvectors and eigenvalues to learn node or edge positional encodings. Mialon et al. [2021] uses spectral kernels such as the diffusion kernel to define relative positional encodings that modulate the attention matrix. Ying et al. [2021] achieve state-of-the-art empirical performance with simple Transformers that incorporate shortest-path based relative positional encodings. Zhang

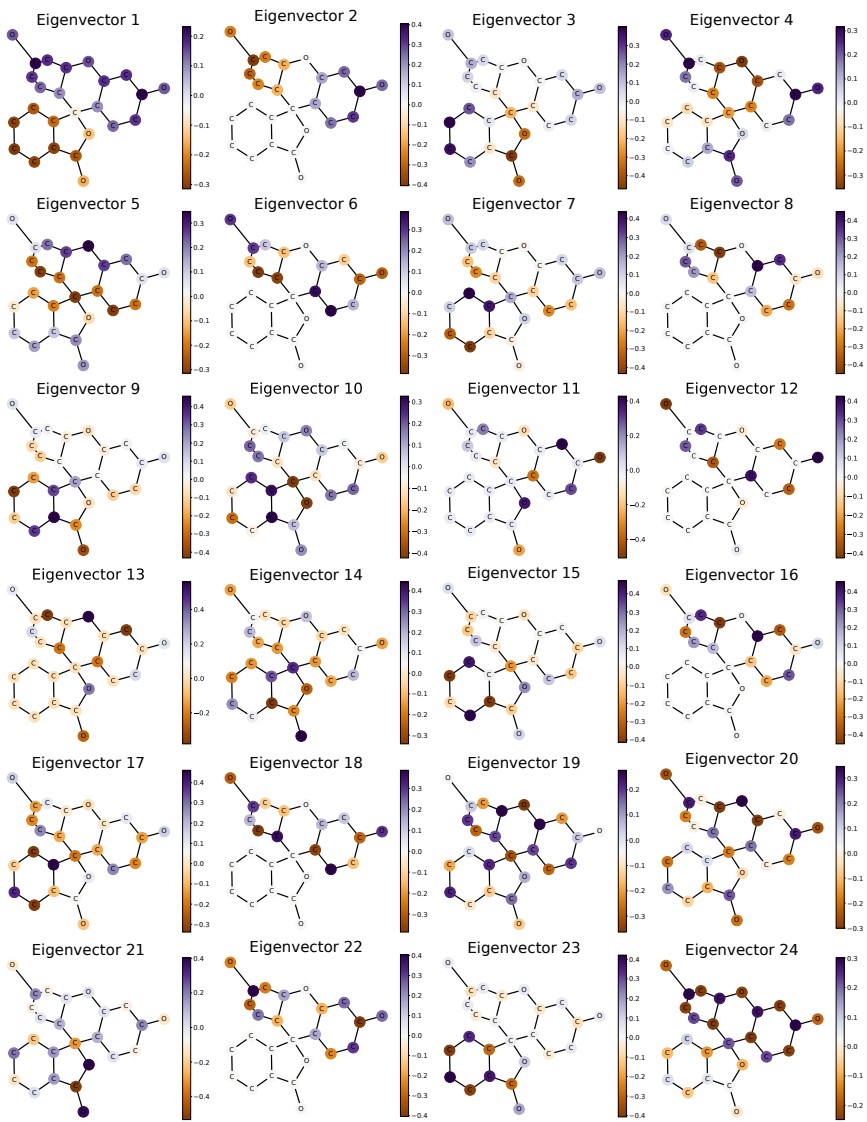

Figure 8: All normalized Laplacian eigenvectors of the fluorescein graph. The first principal components of SignNet's learned positional encodings do not exactly match any eigenvectors.

et al. [2020] also utilize shortest-path distances for positional encodings in their graph Transformer. Kim et al. [2021] develop higher-order transformers (that generalize invariant graph networks), which interestingly perform well on graph regression using sparse higher-order transformers without positional encodings.

## E.2 Eigenvector Symmetries in Graph Representation Learning

Many works that attempt to respect the invariances of eigenvectors solely focus on sign invariance (by using data augmentation) [Dwivedi et al., 2020, Dwivedi and Bresson, 2021, Dwivedi et al., 2022, Kreuzer et al., 2021]. This may be reasonable for continuous data, where eigenvalues of associated matrices may be usually distinct and separated (e.g. Puny et al. [2022] finds that this empirically holds for covariance matrices of $n$-body problems). However, discrete graph Laplacians are known to have higher multiplicity eigenvalues in many cases, and in Appendix C.2 we find this to be true in various types of real-world graph data. Graphs without higher multiplicity eigenspaces are easier to deal with; in fact, graph isomorphism can be tested in polynomial time on graphs of bounded

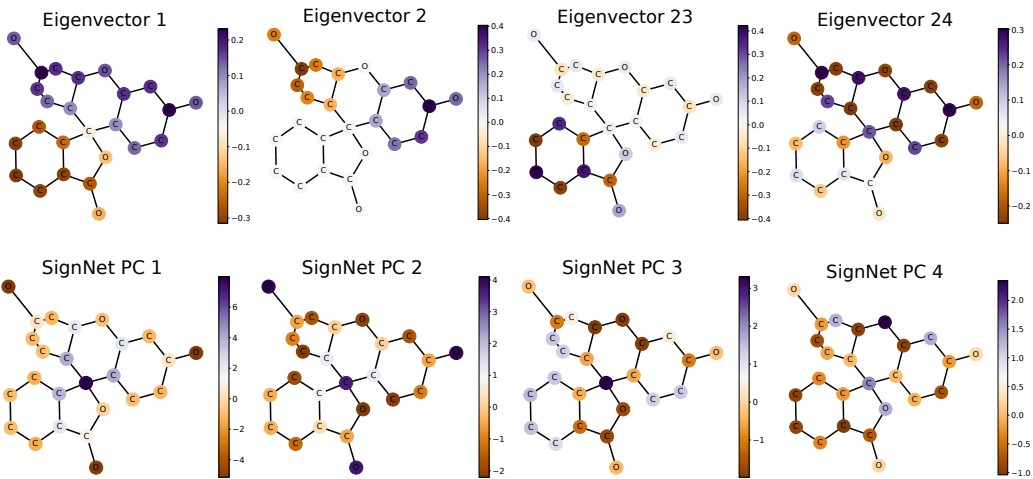

Figure 9: Normalized Laplacian eigenvectors and learned positional encodings for the graph of fluorescein. (Top row) From left to right: smallest and second smallest nontrivial eigenvectors, then second largest and largest eigenvectors. (Bottom row) From left to right: first four principal components of the output $\rho([\phi(v_i) + \phi(-v_i)]_{i=1,\ldots,n})$ of SignNet. Note: we will put this back in the main paper for the camera-ready.

multiplicity for adjacency matrix eigenvalues [Babai et al., 1982], with a time complexity that is lower for graphs with lower maximum multiplicities.

A recent work of Wang et al. [2022] proposes full orthogonal group invariance for functions that process positional encodings. In particular, for positional encodings $Z \in \mathbb{R}^{n \times k}$, they parameterize functions $f(Z)$ such that $f(Z) = f(ZQ)$ for all $Q \in O(k)$. This indeed makes sense for network embeddings like node2vec [Grover and Leskovec, 2016], as their objective functions are based on inner products and are thus orthogonally invariant. While they prove stability results when enforcing full orthogonal invariance for eigenvectors, this is a very strict constraint compared to our basis invariance. For instance, when $k = n$ and all eigenvectors are used in $V$, the condition $f(V) = f(VQ)$ implies that $f$ is a constant function on orthogonal matrices, since any orthogonal matrix $W$ can be obtained as $W = VQ$ for $Q = V^\top W \in O(n)$. In other words, for bases of eigenspaces $V_1, \ldots, V_l$ and $V = [V_1 \quad \ldots \quad V_l]$, Wang et al. [2022] enforces $VQ \cong V$, while we enforce $V \mathrm{Diag}(Q_1, \ldots, Q_l) \cong V$. While the columns of $V \mathrm{Diag}(Q_1, \ldots, Q_l)$ are still eigenvectors, the columns of $VQ$ generally are not.

### E.3 Graph Spectra and Learning on Graphs

More generally, graph spectra are widely used in analyzing graphs, and spectral graph theory [Chung, 1997] studies the connection between graph properties and graph spectra. Different graph kernels have been defined based on graph spectra, which use robust and discriminative notions of generalized spectral distance [Verma and Zhang, 2017], the spectral density of states [Huang et al., 2021], random walk return probabilities [Zhang et al., 2018b], or the trace of the heat kernel [Tsitsulin et al., 2018]. Graph signal processing relies on spectral operations to define Fourier transforms, frequencies, convolutions, and other useful concepts for processing data on graphs [Ortega et al., 2018]. The closely related spectral graph neural networks [Wu et al., 2020, Balcilar et al., 2020] parameterize neural architectures that are based on similar spectral operations.

## F   Definitions, Notation, and Background

### F.1   Basic Topology and Algebra Definitions

We will use some basic topology and algebra for our theoretical results. A topological space $(\mathcal{X}, \tau)$ is a set $\mathcal{X}$ along with a family of subsets $\tau \subseteq 2^{\mathcal{X}}$ satisfying certain properties, which gives useful notions like continuity and compactness. From now on, we will omit mention of $\tau$, and refer to a

topological space as the set $\mathcal{X}$ itself. For topological spaces $\mathcal{X}$ and $\mathcal{Y}$, we write $\mathcal{X} \cong \mathcal{Y}$ and say that $\mathcal{X}$ is homeomorphic to $\mathcal{Y}$ if there exists a continuous bijection with continuous inverse from $\mathcal{X}$ to $\mathcal{Y}$. We will say $\mathcal{X} = \mathcal{Y}$ if the underlying sets and topologies are equal as sets (we will often use this notion of equality for simplicity, even though it can generally be substituted with homeomorphism). For a function $f : \mathcal{X} \to \mathcal{Y}$ between topological spaces $\mathcal{X}$ and $\mathcal{Y}$, the image $\mathrm{im} f$ is the set of values that $f$ takes, $\mathrm{im} f = \{f(x) : x \in \mathcal{X}\}$. This is also denoted $f(\mathcal{X})$. A function $f : \mathcal{X} \to \mathcal{Y}$ is called a topological embedding if it is a homeomorphism from $\mathcal{X}$ to its image.

A group $G$ is a set along with a multiplication operation $G \times G \to G$, such that multiplication is associative, there is a multiplicative identity $e \in G$, and each $g \in G$ has a multiplicative inverse $g^{-1}$. A topological group is a group that is also a topological space such that the multiplication and inverse operations are continuous.

A group $G$ may act on a set $\mathcal{X}$ by a function $\cdot : G \times \mathcal{X} \to \mathcal{X}$. We usually denote $g \cdot x$ as $gx$. A topological group is said to act continuously on a topological space $\mathcal{X}$ if $\cdot$ is continuous. For any group $G$ and topological space $\mathcal{X}$, we define the coset $Gx = \{gx : g \in G\}$, which can be viewed as an equivalance class of elements that can be transformed from one to another by a group element. The quotient space $\mathcal{X}/G = \{Gx : x \in \mathcal{X}\}$ is the set of all such equivalence classes, with a topology induced by that of $\mathcal{X}$. The quotient map $\pi : \mathcal{X} \to \mathcal{X}/G$ is a surjective continuous map that sends $x$ to its coset, $\pi(x) = Gx$.

For $x \in \mathbb{R}^s$, $\|x\|_2$ denotes the standard Euclidean norm. By the $\infty$ norm of functions $f : \mathcal{Z} \to \mathbb{R}^s$ from a compact $\mathcal{Z}$ to a Euclidean space $\mathbb{R}^s$, we mean $\|f\|_\infty = \sup_{z \in \mathcal{Z}} \|f(z)\|_2$.

## F.2 Background on Eigenspace Invariances

Let $V = [v_1 \ \ldots \ v_d]$ and $W = [w_1 \ \ldots \ w_d] \in \mathbb{R}^{n \times d}$ be two orthonormal bases for the same $d$ dimensional subspace of $\mathbb{R}^n$. Since $V$ and $W$ span the same space, their orthogonal projectors are the same, so $VV^\top = WW^\top$. Also, since $V$ and $W$ have orthonormal columns, we have $V^\top V = W^\top W = I \in \mathbb{R}^{d \times d}$. Define $Q = V^\top W$. Then $Q$ is orthogonal because

$$Q^\top Q = W^\top V V^\top W = W^\top W W^\top W = I \tag{19}$$

Moreover, we have that

$$VQ = VV^\top W = WW^\top W = W \tag{20}$$

Thus, for any orthonormal bases $V$ and $W$ of the same subspace, there exists an orthogonal $Q \in O(d)$ such that $VQ = W$.

For another perspective on this, define the Grassmannian $\mathrm{Gr}(d, n)$ as the smooth manifold consisting of all $d$ dimensional subspaces of $\mathbb{R}^n$. Further define the Stiefel manifold $\mathrm{St}(d, n)$ as the set of all orthonormal tuples $[v_1 \ \ldots \ v_d] \in \mathbb{R}^{n \times d}$ of $d$ vectors in $\mathbb{R}^n$. Letting $O(d)$ act by right multiplication, it holds that $\mathrm{St}(d, n)/O(d) \cong \mathrm{Gr}(d, n)$. This implies that any $O(d)$ invariant function on $\mathrm{St}(d, n)$ can be viewed as a function on subspaces. See e.g. Gallier and Quaintance [2020] Chapter 5 for more information on this. We will use this relationship in our proofs of universal representation.

When we consider permutation invariance or equivariance, the permutation acts on dimensions of size $n$. Then a tensor $X \in \mathbb{R}^{n^k \times d}$ is called an order $k$ tensor with respect to this permutation symmetry, where order 0 are called scalars, order 1 tensors are called vectors, and order 2 tensors are called matrices. Note that this does not depend on $d$; in this work, we only ever consider vectors and scalars with respect to the $O(d)$ action.

# G Proofs of Universality

We begin by proving the two propositions for the single subspace case from Section 2.1.

**Proposition 1.** *A continuous function* $h : \mathbb{R}^n \to \mathbb{R}^s$ *is sign invariant if and only if*

$$h(v) = \phi(v) + \phi(-v) \tag{3}$$

*for some continuous* $\phi : \mathbb{R}^n \to \mathbb{R}^s$. *A continuous* $h : \mathbb{R}^n \to \mathbb{R}^n$ *is sign invariant and permutation equivariant if and only if* (3) *holds for a continuous permutation equivariant* $\phi : \mathbb{R}^n \to \mathbb{R}^n$.

*Proof.* If $h(v) = \phi(v) + \phi(-v)$, then $h$ is obviously sign invariant. On the other hand, if $h$ is sign invariant, then letting $\phi(v) = h(v)/2$ gives that $h(v) = \phi(v) + \phi(-v)$, and $\phi$ is of course continuous.

If $h(v) = \phi(v) + \phi(-v)$ for a permutation equivariant $\phi$, then $h(-Pv) = \phi(-Pv) + \phi(Pv) = P\phi(-v) + P\phi(v) = P(\phi(v) + \phi(-v)) = Ph(v)$, so $h$ is permutation equivariant and sign invariant. If $h$ is permutation equivariant and sign invariant, then define $\phi(v) = h(v)/2$ again; it is clear that $\phi$ is continuous and permutation equivariant. $\qquad\square$

**Proposition 2.** *Any continuous, $O(d)$ invariant $h : \mathbb{R}^{n \times d} \to \mathbb{R}^s$ is of the form $h(V) = \phi(VV^\top)$ for a continuous $\phi$. For a compact domain $\mathcal{Z} \subseteq \mathbb{R}^{n \times d}$, maps of the form $V \mapsto \mathrm{IGN}(VV^\top)$ universally approximate continuous functions $h : \mathcal{Z} \subseteq \mathbb{R}^{n \times d} \to \mathbb{R}^n$ that are $O(d)$ invariant and permutation equivariant.*

*Proof.* The case without permutation equivariance holds by the First Fundamental Theorem of $O(d)$ (Lemma 2).

For the permutation equivariant case, let $\mathcal{Z}' = \{VV^\top : V \in \mathcal{Z}\}$ and let $\epsilon > 0$. Note that $\mathcal{Z}'$ is compact, as it is the continuous image of a compact set. Since $h$ is $O(d)$ invariant, the first fundamental theorem of $O(d)$ shows that there exists a continuous function $\phi : \mathcal{Z}' \subseteq \mathbb{R}^{n \times n} \to \mathbb{R}^n$ such that $h(V) = \phi(VV^\top)$. Since $h$ is permutation equivariant, for any permutation matrix $P$ we have that

$$h(PV) = P \cdot h(V) \tag{21}$$

$$\phi(PVV^\top P^\top) = P \cdot \phi(VV^\top), \tag{22}$$

so $\phi$ is a continuous permutation equivariant function from matrices to vectors. Then note that Keriven and Peyré [2019] show that invariant graph networks (of generally high tensor order in hidden layers) universally approximate continuous permutation equivariant functions from matrices to vectors on compact sets of matrices. Thus, an IGN can $\epsilon$-approximate $\phi$, and hence $V \mapsto \mathrm{IGN}(VV^\top)$ can $\epsilon$-approximate $h$. $\qquad\square$

## G.1 Proof of Decomposition Theorem

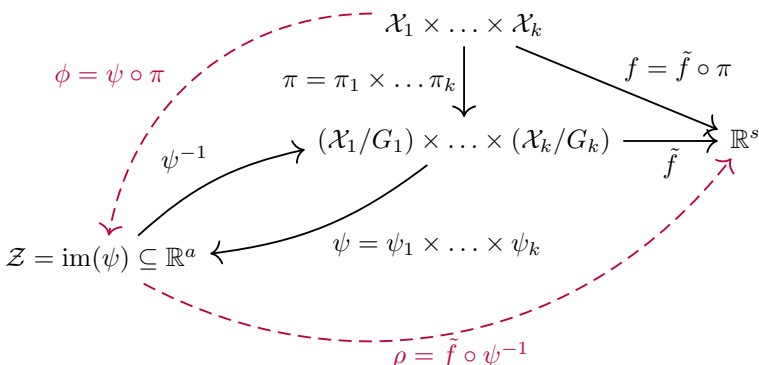

Figure 10: Commutative diagram for our proof of Theorem 3. Black arrows denote functions from topological constructions, and red dashed lines denote functions that we parameterize by neural networks ($\phi = \phi_1 \times \ldots \times \phi_k$ and $\rho$).

Here, we give the formal statement of Theorem 3, which provides the necessary topological assumptions for the theorem to hold. In particular, we only require the $G_i$ be a topological group that acts continuously on $\mathcal{X}_i$ for each $i$, and that there exists a topological embedding of each quotient space into some Euclidean space. That the group action is continuous is a very mild assumption, and it holds for any finite or compact matrix group, which all of the invariances we consider in this paper can be represented as.

A topological embedding of the quotient space into a Euclidean space is desired, as we know how to parameterize neural networks with Euclidean outputs and inputs, whereas dealing with a quotient

space is generally difficult. Many different conditions can guarantee existence of such an embedding. For instance, if the quotient space is a smooth manifold, then the Whitney Embedding Theorem (Lemma 5) guarantees such an embedding. Also, if the base space $\mathcal{X}_i$ is a Euclidean space and $G_i$ is a finite or compact matrix Lie group, then a map built from $G$-invariant polynomials gives such an embedding (González and de Salas [2003] Lemma 11.13).

Figure 10 provides a commutative diagram representing the constructions in our proof.

**Theorem 3** (Decomposition Theorem). *Let $\mathcal{X}_1, \ldots, \mathcal{X}_k$ be topological spaces, and let $G_i$ be a topological group acting continuously on $\mathcal{X}_i$ for each $i$. Assume that there is a topological embedding $\psi_i : \mathcal{X}_i/G_i \to \mathbb{R}^{a_i}$ of each quotient space into a Euclidean space $\mathbb{R}^{a_i}$ for some dimension $a_i$. Then, for any continuous function $f : \mathcal{X} = \mathcal{X}_1 \times \ldots \times \mathcal{X}_k \to \mathbb{R}^s$ that is invariant to the action of $G = G_1 \times \ldots \times G_k$, there exists continuous functions $\phi_i : \mathcal{X}_i \to \mathbb{R}^{a_i}$ and a continuous function $\rho : \mathcal{Z} \subseteq \mathbb{R}^a \to \mathbb{R}^s$, where $a = \sum_i a_i$ such that*

$$f(v_1, \ldots, v_k) = \rho(\phi_1(v_1), \ldots, \phi_k(v_k)). \tag{23}$$

*Furthermore: (1) each $\phi_i$ can be taken to be invariant to $G_i$, (2) the domain $\mathcal{Z}$ is compact if each $\mathcal{X}_i$ is compact, (3) if $\mathcal{X}_i = \mathcal{X}_j$ and $G_i = G_j$, then $\phi_i$ can be taken to be equal to $\phi_j$.*

*Proof.* Let $\pi_i : \mathcal{X}_i \to \mathcal{X}_i/G_i$ denote the quotient map for $\mathcal{X}_i/G_i$. Since each $G_i$ acts continuously, Lemma 3 gives that the quotient of the product space is the product of the quotient spaces, i.e. that

$$(\mathcal{X}_1 \times \ldots \times \mathcal{X}_k)/(G_1 \times \ldots G_k) \cong (\mathcal{X}_1/G_1) \times \ldots \times (\mathcal{X}_k/G_k), \tag{24}$$

and the corresponding quotient map $\pi : \mathcal{X}/G$ is given by

$$\pi = \pi_1 \times \ldots \times \pi_k, \qquad \pi(x_1, \ldots, x_k) = (\pi_1(x_1), \ldots, \pi_k(x_k)). \tag{25}$$

By passing to the quotient (Lemma 1), there exists a continuous $\tilde{f} : \mathcal{X}/G \to \mathbb{R}^s$ on the quotient space such that $f = \tilde{f} \circ \pi$. By Lemma 4, each $\mathcal{X}_i/G_i$ is compact if $\mathcal{X}_i$ is compact. Defining the image $\mathcal{Z}_i = \psi_i(\mathcal{X}_i/G_i) \subseteq \mathbb{R}^{a_i}$, we thus know that $\mathcal{Z}_i$ is compact if $\mathcal{X}_i$ is compact.

Moreover, as $\psi_i$ is a topological embedding, it has a continuous inverse $\psi_i^{-1}$ on its image $\mathcal{Z}_i$. Further, we have a topological embedding $\psi : \mathcal{X}/G \to \mathcal{Z} = \mathcal{Z}_1 \times \ldots \times \mathcal{Z}_k$ given by $\psi = \psi_1 \times \ldots \times \psi_k$, with continuous inverse $\psi^{-1} = \psi_1^{-1} \times \ldots \times \psi_k^{-1}$.

Note that

$$f = \tilde{f} \circ \pi = (\tilde{f} \circ \psi^{-1}) \circ (\psi \circ \pi). \tag{26}$$

So we define

$$\rho = \tilde{f} \circ \psi^{-1} \qquad\qquad \rho : \mathcal{Z} \to \mathbb{R}^s \tag{27}$$
$$\phi_i = \psi_i \circ \pi_i \qquad\qquad \phi_i : \mathcal{X}_i \to \mathcal{Z}_i \tag{28}$$
$$\phi = \psi \circ \pi = \phi_1 \times \ldots \times \phi_k \qquad\qquad \phi : \mathcal{X} \to \mathcal{Z} \tag{29}$$

Thus, $f = \rho \circ \phi = \rho \circ (\phi_1 \times \ldots \times \phi_k)$, so equation (9) holds. Moreover, the $\rho$ and $\phi_i$ are continuous, as they are compositions of continuous functions. Furthermore, (1) holds as each $\phi_i$ is invariant to $G_i$ because each $\pi_i$ is invariant to $G_i$. Since each $\mathcal{Z}_i$ is compact if $\mathcal{X}_i$ is compact, the product $\mathcal{Z} = \mathcal{Z}_1 \times \ldots \times \mathcal{Z}_k$ is compact if each $\mathcal{X}_i$ is compact, thus proving (2).

To show the last statement (3), note simply that if $\mathcal{X}_i = \mathcal{X}_j$ and $G_i = G_j$, then the quotient maps are equal, i.e. $\pi_i = \pi_j$. Moreover, we can choose the embeddings to be equal, so say $\psi_i = \psi_j$. Then, $\phi_i = \psi_i \circ \pi_i = \psi_j \circ \pi_j = \phi_j$, so we are done. □

## G.2 Universality of SignNet and BasisNet

Here, we prove Corollary 1 on the universal representation and approximation capabilities of our Unconstrained-SignNets, Unconstrained-BasisNets, and Expressive-BasisNets. We proceed in several steps, first proving universal representation of continuous functions when we do not require permutation equivariance, then proving universal approximation when we do require permutation equivariance.

 **G.2.1   Sign Invariant Universal Representation**

Recall that $\mathbb{S}^{n-1}$ denotes the unit sphere in $\mathbb{R}^n$. As we normalize eigenvectors to unit norm, the domain of our functions on $k$ eigenvectors are on the compact space $(\mathbb{S}^{n-1})^k$.

**Corollary 2** (Universal Representation for SignNet). *A continuous function $f : (\mathbb{S}^{n-1})^k \to \mathbb{R}^s$ is sign invariant, i.e. $f(s_1 v_1, \ldots, s_k v_k) = f(v_1, \ldots, v_k)$ for any $s_i \in \{-1, 1\}$, if and only if there exists a continuous $\phi : \mathbb{R}^n \to \mathbb{R}^{2n-2}$ and a continuous $\rho : \mathbb{R}^{(2n-2)k} \to \mathbb{R}^s$ such that*

$$f(v_1, \ldots, v_k) = \rho\left([\phi(v_i) + \phi(-v_i)]_{i=1}^k\right). \tag{30}$$

*Proof.* It can be directly seen that any $f$ of the above form is sign invariant.

Thus, we show that any sign invariant $f$ can be expressed in the above form. First, we show that we can apply the general Theorem 3. The group $G_i = \{1, -1\}$ acts continuously and satisfies that $\mathbb{S}^{n-1}/\{1, -1\} = \mathbb{RP}^{n-1}$, where $\mathbb{RP}^{n-1}$ is the real projective space of dimension $n-1$. Since $\mathbb{RP}^{n-1}$ is a smooth manifold of dimension $n-1$, Whitney's embedding theorem states that there exists a (smooth) topological embedding $\psi_i : \mathbb{RP}^{n-1} \to \mathbb{R}^{2n-2}$ (Lemma 5).

Thus, we can apply the general theorem to see that $f = \rho \circ \tilde{\phi}^k$ for some continuous $\rho$ and $\tilde{\phi}^k$. Note that each $\tilde{\phi}_i = \tilde{\phi}$ is the same, as each $\mathcal{X}_i = \mathbb{S}^{n-1}$ and $G_i = \{1, -1\}$ is the same. Also, Theorem 3 says that we may assume that $\tilde{\phi}$ is sign invariant, so $\tilde{\phi}(x) = \tilde{\phi}(-x)$. Letting $\phi(x) = \tilde{\phi}(x)/2$, we are done with the proof. $\qquad\square$

**G.2.2   Sign Invariant Universal Representation with Extra Features**

Recall that we may want our sign invariant functions to process other data besides eigenvectors, such as eigenvalues or node features associated to a graph. Here, we show universal representation for when we have this other data that does not possess sign symmetry. The proof is a simple extension of Corollary 2, but we provide the technical details for completeness.

**Corollary 3** (Universal Representation for SignNet with features). *For a compact space of features $\Omega \subseteq \mathbb{R}^d$, let $f(v_1, \ldots, v_k, x_1, \ldots, x_k)$ be a continuous function $f : (\mathbb{S}^{n-1} \times \Omega)^k \to \mathbb{R}^s$.*

*Then $f$ is sign invariant for the inputs on the sphere, i.e.*

$$f(s_1 v_1, \ldots, s_k v_k, x_1, \ldots, x_k) = f(v_1, \ldots, v_k, x_1, \ldots, x_k) \qquad s_i \in \{1, -1\}, \tag{31}$$

*if and only if there exists a continuous $\psi : \mathbb{R}^{n+d} \to \mathbb{R}^{2n-2+d}$ and a continuous $\rho : \mathbb{R}^{(2n-2+d)k} \to \mathbb{R}^s$ such that*

$$f(v_1, \ldots, v_k) = \rho\left(\phi(v_1, x_1) + \phi(-v_1, x_1), \ldots, \phi(v_k, x_k) + \phi(-v_k, x_k)\right). \tag{32}$$

*Proof.* Once again, the sign invariance of any $f$ in the above form is clear.

We follow very similar steps to the proof of Corollary 2 to show that we may apply Theorem 3. We can view $\Omega$ as a quotient space, after quotienting by the trivial group that does nothing, $\Omega \cong \Omega/\{1\}$. The corresponding quotient map is $\mathrm{id}_\Omega$, the identity map. Also, $\Omega$ trivially topologically embeds in $\mathbb{R}^d$ by the inclusion map.

As $G_i = \{-1, 1\} \times \{1\}$ acts continuously, by Lemma 3 we have that

$$(\mathbb{S}^{n-1} \times \Omega)/(\{1, -1\} \times \{1\}) \cong (\mathbb{S}^{n-1}/\{1, -1\}) \times (\Omega/\{1\}) \cong \mathbb{RP}^{n-1} \times \Omega, \tag{33}$$

with corresponding quotient map $\pi \times \mathrm{id}_\Omega$, where $\pi$ is the quotient map to $\mathbb{RP}^{n-1}$.

Letting $\tilde{\psi}$ be the embedding of $\mathbb{RP}^{n-1} \to \mathbb{R}^{2n-2}$ guaranteed by Whitney's embedding theorem (Lemma 5), we have that $\psi = \tilde{\psi} \times \mathrm{id}_\Omega$ is an embedding of $\mathbb{RP}^{n-1} \times \Omega \to \mathbb{R}^{2n-2+d}$. Thus, we can apply Theorem 3 to write $f = \rho \circ \tilde{\phi}^k$ for $\tilde{\phi} = (\tilde{\psi} \times \mathrm{id}_\Omega) \circ (\pi \times \mathrm{id}_\Omega)$, so

$$\tilde{\phi}(v_i, x_i) = (\tilde{\psi}(v_i), x_i), \tag{34}$$

where $\tilde{\phi}(v_i, x_i) = \tilde{\phi}(-v_i, x_i)$. Letting $\phi(v_i, x_i) = \tilde{\phi}(v_i, x_i)/2$, we are done. $\qquad\square$

### G.2.3  Basis Invariant Universal Representation

Recall that $\mathrm{St}(d, n)$ is the Stiefel manifold of $d$-tuples of vectors $(v_1, \ldots, v_d)$ where $v_i \in \mathbb{R}^n$ and $v_1, \ldots, v_d$ are orthonormal. This is where our inputs lie, as our eigenvectors are unit norm and orthogonal. We will also make use of the Grassmannian $\mathrm{Gr}(d, n)$, which consists of all $d$-dimensional subspaces in $\mathbb{R}^n$. This is because the Grassmannian is the quotient space for the group action we want, $\mathrm{Gr}(d, n) \cong \mathrm{St}(d, n)/O(d)$, where $Q \in O(d)$ acts on $V \in \mathrm{St}(d, n) \subseteq \mathbb{R}^{n \times d}$ by mapping $V$ to $VQ$ [Gallier and Quaintance, 2020].

**Corollary 4** (Universal Representation for BasisNet). *For dimensions $d_1, \ldots, d_l \leq n$ let $f$ be a continuous function on $\mathrm{St}(d_1, n) \times \ldots \times \mathrm{St}(d_l, n)$. Further assume that $f$ is invariant to $O(d_1) \times \ldots \times O(d_l)$, where $O(d_i)$ acts on $\mathrm{St}(d_i, n)$ by multiplication on the right.*

*Then there exist continuous $\rho : \mathbb{R}^{\sum_{i=1}^{l} 2 d_i(n-d_i)} \to \mathbb{R}^s$ and continuous $\phi_i : \mathrm{St}(d_i, n) \to \mathbb{R}^{2 d_i(n-d_i)}$ such that*

$$f(V_1, \ldots, V_l) = \rho\left(\phi_1(V_1), \ldots, \phi_l(V_l)\right), \tag{35}$$

*where the $\phi_i$ are $O(d_i)$ invariant functions, and we can take $\phi_i = \phi_j$ if $d_i = d_j$.*

*Proof.* Letting $\mathcal{X}_i = \mathrm{St}(d_i, n)$ and $G_i = O(d_i)$, it can be seen that $G_i$ acts continuously on $\mathcal{X}_i$. Also, we have that the quotient space $\mathrm{St}(d_i, n)/O(d_i) = \mathrm{Gr}(d_i, n)$ is the Grassmannian of $d_i$ dimensional subspaces in $\mathbb{R}^n$, which is a smooth manifold of dimension $d_i(n - d_i)$. Thus, the Whitney embedding theorem (Lemma 5) gives a topological embedding $\psi_i : \mathrm{Gr}(d_i, n) \to \mathbb{R}^{2 d_i(n-d_i)}$.

Hence, we may apply Theorem 3 to obtain continuous $O(d_i)$ invariant $\phi_i : \mathrm{St}(d_i, n) \to \mathbb{R}^{2 d_i(n-d_i)}$ and continuous $\rho : \mathbb{R}^{\sum_{i=1}^{l} 2 d_i(n-d_i)} \to \mathbb{R}^s$, such that $f = \rho \circ (\phi_1 \times \ldots \times \phi_l)$. Also, if $d_i = d_j$, then $\mathcal{X}_i = \mathcal{X}_j$ and $G_i = G_j$, so we can take $\phi_i = \phi_j$.

$\square$

### G.2.4  Basis Invariant and Permutation Equivariant Universal Approximation

With the restriction that $f(V_1, \ldots, V_l) : \mathbb{R}^{n \times \sum_i d_i} \to \mathbb{R}^n$ be permutation equivariant and basis invariant, we need to use the impractically expensive Expressive-BasisNet to approximate $f$. Universality of permutation invariant or equivariant functions from matrices to scalars or matrices to vectors is difficult to achieve in a computationally tractable manner [Maron et al., 2019, Keriven and Peyré, 2019, Maehara and NT, 2019]. One intuitive reason to expect this is that universally approximating such functions allows solution of the graph isomorphism problem [Chen et al., 2019b], which is a computationally difficult problem. While we have exact representation of basis invariant functions by continuous $\rho$ and $\phi_i$ when there is no permutation equivariance constraint, we can only achieve approximation up to an arbitrary $\epsilon > 0$ when we require permutation equivariance.

**Corollary 5** (Universal Approximation for Expressive-BasisNets). *Let $f(V_1, \ldots, V_l) : \mathrm{St}(d_1, n) \times \ldots \times \mathrm{St}(d_l, n) \to \mathbb{R}^n$ be continuous, $O(d_1) \times \ldots \times O(d_l)$ invariant, and permutation equivariant. Then $f$ can be $\epsilon$-approximated by an Expressive-BasisNet.*

*Proof.* By invariance, Corollary 4 of the decomposition theorem shows that $f$ can be written as

$$f(V_1, \ldots, V_l) = \rho\left(\varphi_{d_1}(V_1), \ldots, \varphi_{d_l}(V_l)\right) \tag{36}$$

for some continuous $O(d_i)$ invariant $\varphi_{d_i}$ and continuous $\rho$. By the first fundamental theorem of $O(d)$ (Lemma 2), each $\varphi_{d_i}$ can be written as $\varphi_{d_i}(V_i) = \phi_{d_i}(V_i V_i^\top)$ for some continuous $\phi_{d_i}$. Let

$$\mathcal{Z} = \{(V_1 V_1^\top, \ldots, V_l V_l^\top) : V_i \in \mathrm{St}(d_i, n)\} \subseteq \mathbb{R}^{n^2 \times l}, \tag{37}$$

which is compact as it is the image of the compact space $\mathrm{St}(d_1, n) \times \ldots \times \mathrm{St}(d_l, n)$ under a continuous function. Define $h : \mathcal{Z} \subseteq \mathbb{R}^{n^2 \times l} \to \mathbb{R}^n$ by

$$h(V_1 V_1^\top, \ldots, V_l V_l^\top) = \rho\left(\phi_{d_1}(V_1 V_1^\top), \ldots, \phi_{d_l}(V_l V_l^\top)\right). \tag{38}$$

Then note that $h$ is continuous and permutation equivariant from matrices to vectors, so it can be $\epsilon$-approximated by an invariant graph network [Keriven and Peyré, 2019], call it $\widetilde{\mathrm{IGN}}$. If we define $\tilde{\rho} = \widetilde{\mathrm{IGN}}$ and $\mathrm{IGN}_{d_i}(V_i V_i^\top) = V_i V_i^\top$ (this identity operation is linear and permutation equivariant, so it can be exactly expressed by an IGN), then we have $\epsilon$-approximation of $f$ by

$$\widetilde{\mathrm{IGN}}(V_1 V_1^\top, \ldots, V_l V_l^\top) = \tilde{\rho}\left(\mathrm{IGN}_{d_1}(V_1 V_1^\top), \ldots, \mathrm{IGN}_{d_l}(V_l V_l^\top)\right). \tag{39}$$

$\square$

 **G.3 Proof of Universal Approximation for General Decompositions**

**Theorem 4.** *Consider the same setup as Theorem 3, where $\mathcal{X}_i$ are also compact. Let $\Phi_i$ be a family of $G_i$-invariant functions that universally approximate $G_i$-invariant continuous functions $\mathcal{X}_i \to \mathbb{R}^{a_i}$, and let $\mathcal{R}$ be a set of continuous function that universally approximate continuous functions $\mathcal{Z} \subseteq \mathbb{R}^a \to \mathbb{R}^s$ for every compact $\mathcal{Z}$, where $a = \sum_i a_i$. Then for any $\varepsilon > 0$ and any $G$-invariant continuous function $f : \mathcal{X}_1 \times \ldots \times \mathcal{X}_k \to \mathbb{R}^s$ there exists $\phi \in \Phi$ and $\rho \in \mathcal{R}$ such that $\|f - \rho(\phi_1, \ldots, \phi_k)\|_\infty < \varepsilon$.*

*Proof.* Consider a particular $G$-invariant continuous function $f : \mathcal{X}_1 \times \ldots \times \mathcal{X}_k \to \mathbb{R}^s$. By Theorem 3 there exists $G_i$-invariant continuous functions $\phi_i' : \mathcal{X}_i \to \mathbb{R}^{a_i}$ and a continuous function $\rho' : \mathcal{Z} \subseteq \mathbb{R}^a \to \mathbb{R}^s$ (where $a = \sum_i a_i$) such that

$$f(v_1, \ldots, v_k) = \rho'(\phi_1'(v_1), \ldots, \phi_k'(v_k)).$$

Now fix an $\varepsilon > 0$. For any $\rho \in \mathcal{R}$ and any $\phi_i \in \Phi_i$ ($i = 1, \ldots k$) we may bound the difference from $f$ as follows (suppressing the $v_i$'s for brevity),

$$\begin{aligned}
&\|f - \rho(\phi_1, \ldots, \phi_k)\|_\infty \\
&= \|\rho'(\phi_1', \ldots, \phi_k') - \rho(\phi_1, \ldots, \phi_k)\|_\infty \\
&= \|\rho'(\phi_1', \ldots, \phi_k') - \rho(\phi_1', \ldots, \phi_k') + \rho(\phi_1', \ldots, \phi_k') - \rho(\phi_1, \ldots, \phi_k)\|_\infty \\
&\leq \|\rho'(\phi_1', \ldots, \phi_k') - \rho(\phi_1', \ldots, \phi_k')\|_\infty + \|\rho(\phi_1', \ldots, \phi_k') - \rho(\phi_1, \ldots, \phi_k)\|_\infty \\
&= \text{I} + \text{II}
\end{aligned}$$

Now let $K' = \prod_{i=1}^k \text{im}\phi_i'$. Since each $\phi_i'$ is continuous and defined on a compact set $\mathcal{X}_i$ we know that $\text{im}\phi_i'$ is compact, and so the product $K$ is also compact. Since $K'$ is compact, it is contained in a closed ball $B(r)$ of radius $r > 0$ centered at the origin. Let $K$ be the closed ball $B(r + 1)$ of radius $r + 1$ centered at the origin, so $K$ contains $K'$ and a ball of radius 1 around each point of $K'$. We may extend $\rho'$ continuously to $K$ as needed, so assume $\rho' : K \to \mathbb{R}^s$. By universality of $\mathcal{R}$ we may pick a particular $\rho : K \to \mathbb{R}^s$, $\rho \in \mathcal{R}$ such that

$$\text{I} = \sup_{\{v_i \in \mathcal{X}_i\}_{i=1}^k} \|\rho'(\phi_1', \ldots, \phi_k') - \rho(\phi_1', \ldots, \phi_k')\|_\infty \leq \sup_{z \in K} \|\rho'(z) - \rho(z)\|_2 < \varepsilon/2.$$

Keeping this choice of $\rho$, it remains only to bound II. As $\rho$ is continuous on a compact domain, it is in fact uniformly continuous. Thus, we can choose a $\delta' > 0$ such that if $\|y - z\|_2 \leq \delta'$, then $\|\rho(y) - \rho(z)\|_\infty < \epsilon/2$, and then we define $\delta = \min(\delta', 1)$.

Since $\Phi_i$ universally approximates $\phi_i'$ we may pick $\phi_i \in \Phi_i$ such that $\|\phi_i - \phi_i'\|_\infty < \delta/\sqrt{k}$, and thus $\|(\phi_1, \ldots, \phi_k) - (\phi_1', \ldots \phi_k')\|_\infty \leq \delta$. With this choice of $\phi_i$, we know that $\prod_{i=1}^k \text{im}\phi_i \subseteq K$ (because each $\phi_i(x_i)$ is within distance 1 of $\phi_i'(x_i)$). Thus, $\rho(\phi_1(x_1), \ldots, \phi_k(x_k))$ is well-defined, and we have

$$\begin{aligned}
\text{II} &= \|\rho(\phi_1', \ldots, \phi_k') - \rho(\phi_1, \ldots, \phi_k)\|_\infty \\
&= \sup_{\{x_i \in \mathcal{X}_i\}_{i=1}^k} \|\rho(\phi_1'(x_1), \ldots, \phi_k'(x_k)) - \rho(\phi_1(x_1), \ldots, \phi_k(x_k))\|_2 \\
&< \varepsilon/2
\end{aligned}$$

due to our choice of $\delta$, which completes the proof. $\qquad\square$

# H  Basis Invariance for Graph Representation Learning

## H.1  Spectral Graph Convolution

In this section, we consider spectral graph convolutions, which for node features $X \in \mathbb{R}^{n \times q}$ take the form $f(V, \Lambda, X) = \sum_{i=1}^n \theta_i v_i v_i^\top X$ for some parameters $\theta_i$. We can optionally take $\theta_i = h(\lambda_i)$ for some continuous function $h : \mathbb{R} \to \mathbb{R}$ of the eigenvalues. This form captures most popular spectral graph convolutions in the literature [Bruna et al., 2014, Hamilton, 2020, Bronstein et al., 2017]; often, such convolutions are parameterized by taking $h$ to be some analytic function such as a simple affine function [Kipf and Welling, 2017], a linear combination in a polynomial basis [Defferrard et al.,

2016, Chien et al., 2021], or a parameterization of rational functions [Levie et al., 2018, Bianchi et al., 2021].

First, it is well known and easy to see that spectral graph convolutions are permutation equivariant, as for a permutation matrix $P$ we have

$$f(PV, \Lambda, PX) = \sum_i \theta_i P v_i v_i^\top P^\top P X = \sum_i \theta_i P v_i v_i^\top X = P f(V, \Lambda, X). \tag{40}$$

Also, it is easy to see that they are sign invariant, as $(-v_i)(-v_i)^\top = v_i v_i^\top$. However, if the $\theta_i$ do not depend on the eigenvalues, then the spectral graph convolution is not necessarily basis invariant. For instance, if $v_1$ and $v_2$ are in the same eigenspace, and we change basis by permuting $v_1' = v_2$ and $v_2' = v_1$, then if $\theta_1 \neq \theta_2$ the spectral graph convolution will generally change as well.

On the other hand, if $\theta_i = h(\lambda_i)$ for some function $h : \mathbb{R} \to \mathbb{R}$, then the spectral graph convolution is basis invariant. This is because if $v_i$ and $v_j$ belong to the same eigenspace, then $\lambda_i = \lambda_j$ so $h(\lambda_i) = h(\lambda_j)$. Thus, if $v_{i_1}, \ldots, v_{i_d}$ are eigenvectors of the same eigenspace with eigenvalue $\lambda$, we have that $\sum_{l=1}^d h(\lambda_{i_l}) v_{i_l} v_{i_l}^\top = h(\lambda) \sum_{l=1}^d v_{i_l} v_{i_l}^\top$. Now, note that $\sum_{l=1}^d v_{i_l} v_{i_l}^\top$ is the orthogonal projector onto the eigenspace [Trefethen and Bau III, 1997]. A change of basis does not change this orthogonal projector, so such spectral graph convolutions are basis invariant.

Another way to see this basis invariance is with a simple computation. Let $V_1, \ldots, V_l$ be the eigenspaces of dimension $d_1, \ldots, d_l$, where $V_i \in \mathbb{R}^{n \times d_i}$. Let the corresponding eigenvalues be $\mu_1, \ldots, \mu_l$. Then for any orthogonal matrices $Q_i \in O(d_i)$, we have

$$\sum_{i=1}^n h(\lambda_i) v_i v_i^\top = \sum_{j=1}^l V_j h(\mu_j) I_{d_j} V_j^\top \tag{41}$$

$$= \sum_{j=1}^l V_j h(\mu_j) I_{d_j} Q_j Q_j^\top V_j^\top \tag{42}$$

$$= \sum_{j=1}^l (V_j Q_j) h(\mu_j) I_{d_j} (V_j Q_j)^\top, \tag{43}$$

so the spectral graph convolution is invariant to substituting $V_j Q_j$ for $V_j$.

Now, we give the proof that shows SignNet and BasisNet can universally approximate spectral graph convolutions.

**Theorem 1** (Learning Spectral Graph Convolutions). *Suppose the node features $X \in \mathbb{R}^{n \times q}$ take values in compact sets. Then SignNet can universally approximate any spectral graph convolution, and both BasisNet and Expressive-BasisNet can universally approximate any parametric spectral graph convolution.*

*Proof.* Note that eigenvectors and eigenvalues of normalized Laplacian matrices take values in compact sets, since the eigenvalues are in $[0, 2]$ and we take eigenvectors to have unit-norm. Thus, the whole domain of the spectral graph convolution is compact.

Let $\varepsilon > 0$. First, consider a spectral graph convolution $f(V, \Lambda, X) = \sum_{i=1}^n \theta_i v_i v_i^\top X$. For SignNet, let $\phi(v_i, \lambda_i, X)$ approximate the function $\tilde{\phi}(v_i, \lambda_i, X) = \theta_i v_i v_i^\top X$ to within $\varepsilon/n$ error, which DeepSets can do since this is a continuous permutation equivariant function from vectors to vectors [Segol and Lipman, 2019] (note that we can pass $\lambda_i$ as a vector in $\mathbb{R}^n$ by instead passing $\lambda_i \mathbf{1}$, where $\mathbf{1}$ is the all ones vector). Then $\rho = \sum_{i=1}^n$ is a linear permutation equivariant operation that can be exactly expressed by DeepSets, so the total error is within $\varepsilon$. The same argument applies when $\theta_i = h(\lambda_i)$ for some continuous function $h$.

For the basis invariant case, consider a parametric spectral graph convolution $f(V, \Lambda, X) = \sum_{i=1}^n h(\lambda_i) v_i v_i^\top X$. Note that if the eigenspace bases are $V_1, \ldots, V_l$ with eigenvalues $\mu_1, \ldots, \mu_l$, we can write the $f(V, \Lambda, X) = \sum_{i=1}^l h(\mu_j) V_j V_j^\top X$. Again, we will let $\rho = \sum_{i=1}^l$ be a sum function, which can be expressed exactly by DeepSets. Thus, it suffices to show that $h(\mu_j) V_j V_j^\top X$ can be $\epsilon/n$ approximated by a 2-IGN (i.e. an IGN that only uses vectors and matrices).

Note that since $h$ is continuous, we can use an elementwise MLP (which IGNs can learn) to approximate $f_1(\mu \mathbf{1}\mathbf{1}^\top, VV^\top, X) = (h(\mu)\mathbf{1}\mathbf{1}^\top, VV^\top, X)$ to arbitrary precision (note that we represent the eigenvalue $\mu$ as a constant matrix $\mu \mathbf{1}\mathbf{1}^\top$). Also, since a 2-IGN can learn matrix vector multiplication (Cai and Wang [2022] Lemma 10), we can approximate $f_2(h(\mu)\mathbf{1}\mathbf{1}^\top, VV^\top, X) = (h(\mu)\mathbf{1}\mathbf{1}^\top, VV^\top X)$, as $V_i V_i^\top \in \mathbb{R}^{n^2}$ is a matrix and $X \in \mathbb{R}^{n \times q}$ is a vector with respect to permutation symmetries. Finally, we use an elementwise MLP to approximate the scalar-vector multiplication $f_3(h(\mu)\mathbf{1}\mathbf{1}^\top, VV^\top, X) = h(\mu)VV^\top X$. Since $f_3 \circ f_2 \circ f_1(\mu \mathbf{1}\mathbf{1}^\top, VV^\top, X) = h(\mu)VV^\top X$, and since 2-IGNs universally approximate each $f_i$, applying Lemma 6 shows that a 2-IGN can approximate $h(\mu)VV^\top X$ to $\epsilon/n$ accuracy, so we are done. Since Expressive-BasisNet is stronger than BasisNet, it can also universally approximate these functions. $\square$

From the proof, we can see that SignNet and BasisNet need only learn simple functions for the $\rho$ and $\phi$ when $h$ is simple, or when the filter is non-parametric and we need only learn $\theta_i$. Xu et al. [2020] propose the principle of algorithmic alignment, and show that if separate modules of a neural network each need only learn simple functions (that is, functions that are well-approximated by low-order polynomials with small coefficients), then the network may be more sample efficient. If we do not require permutation equivariance, and parameterize SignNet and BasisNet with simple MLPs, then algorithmic alignment may suggest that our models are sample efficient. Indeed, $\rho = \sum$ is a simple linear function with coefficients 1, and $\phi(V, \lambda, X) = h(\lambda)VV^\top X$ is quadratic in $V$ and linear in $X$, so it is simple if $h$ is simple.

**Proposition 3.** *There exist infinitely many pairs of non-isomorphic graphs that SignNet and BasisNet can distinguish, but spectral graph convolutions or spectral GNNs cannot distinguish.*

*Proof.* The idea is as follows: we will take graphs $G$ and give them the node feature matrix $X_G = D^{1/2}\mathbf{1}$, i.e. each node has as feature the square root of its degree. Then any spectral graph convolution (or, the first layer of any spectral GNN) will map $V \mathrm{Diag}(\theta)V^\top X$ to something that only depends on the degree sequence and number of nodes. Thus, any spectral graph convolution or spectral GNN will have the same output (up to permutation) for any such graphs $G$ with node features $X_G$ and the same number of nodes and same degree sequence. On the other hand, SignNet and BasisNet can distinguish between infinitely many pairs of graphs $(G^{(1)}, G^{(2)})$ with node features $(X_{G^{(1)}}, X_{G^{(2)}})$ and the same number of nodes and degree sequence; this is because SignNet and BasisNet can tell when a graph is bipartite.

For each $n \geq 5$, we will define $G^{(1)}$ and $G^{(2)}$ as connected graphs with $n$ nodes, with the same degree sequence. Also, we define $G^{(1)}$ to have node features $X_i^{(1)} = \sqrt{d_i^{(1)}}$, where $d_i^{(1)}$ is the degree of node $i$ in $G^{(1)}$, and similarly $G^{(2)}$ has node features $X_i^{(2)} = \sqrt{d_i^{(2)}}$. Now, note that $X^{(1)}$ is an eigenvector of the normalized Laplacian of $G^{(1)}$, and it has eigenvalue 0. As we take the eigenvectors to be orthonormal (since the normalized Laplacian is symmetric), for any spectral graph convolution we have that

$$\sum_{i=1}^{n} \theta_i v_i v_i^\top X^{(1)} = \theta_1 v_1 v_1^\top X^{(1)} = \theta_1 D_1^{1/2}\mathbf{1}(D_1^{1/2}\mathbf{1})^\top D_1^{1/2}\mathbf{1} = \theta_1 \sum_{j=1}^{n}(d_j^{(1)})D_1^{1/2}\mathbf{1}. \qquad (44)$$

Where $D_1$ is the diagonal degree matrix of $G^{(1)}$. Likewise, any spectral graph convolution outputs $\theta_1 \sum_j (d_j^{(2)})D_2^{1/2}\mathbf{1}$ for $G^{(2)}$. Since $D_1$ and $D_2$ are the same up to a permutation, we have that any spectral graph convolution has the same output for $G^{(1)}$ and $G^{(2)}$, up to a permutation. In fact, this also holds for spectral GNNs, as the first layer will always have the same output (up to a permutation) on $G^{(1)}$ and $G^{(2)}$, so the latter layers will also have the same output up to a permutation.

Now, we concretely define $G^{(1)}$ and $G^{(2)}$. This is illustrated in Figure 11 and Figure 12. For $n = 5$, let $G^{(1)}$ contain a triangle with nodes $w_1, w_2, w_3$, and have a path of length 2 coming out of one of the nodes in the triangle, say $w_1$ connects to $w_4$, and $w_4$ connects to $w_5$. This is not bipartite, as there is a triangle. Let $G^{(2)}$ be a bipartite graph that has 2 nodes on the left $(v_1, v_2)$ and 3 nodes on the right $(v_3, v_4, v_5)$. Connect $v_1$ with all nodes on the right, and connect $v_2$ with $v_3$ and $v_4$.

Note that both $G^{(1)}$ and $G^{(2)}$ have the same number of nodes and the same degree sequence $\{3, 2, 2, 2, 1\}$. Thus, spectral graph convolutions or spectral GNNs cannot distinguish them. How-

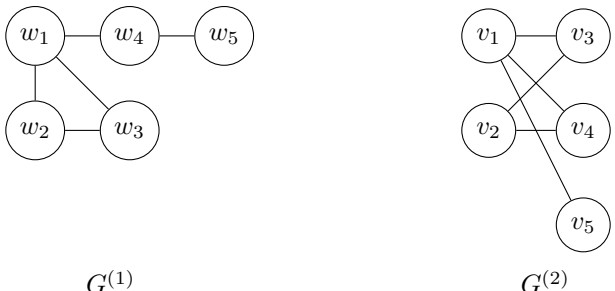

Figure 11: Illustration of our constructed $G^{(1)}$ and $G^{(2)}$ for $n = 5$, as used in the proof of Proposition 3.

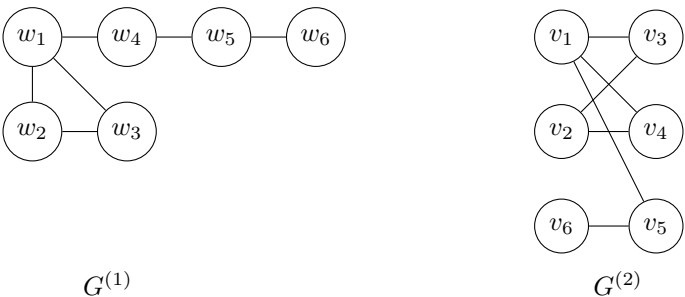

Figure 12: Illustration of our constructed $G^{(1)}$ and $G^{(2)}$ for $n = 6$, as used in the proof of Proposition 3.

ever, SignNet and BasisNet can distinguish them, as they can tell whether a graph is bipartite by checking the highest eigenvalue of the normalized Laplacian. This is because the multiplicity of the eigenvalue 2 is the number of bipartite components. In particular, SignNet can approximate the function $\phi(v_i, \lambda_i, X) = \lambda_i$ and $\rho \approx \max_{i=1}^{n}$. Likewise, BasisNet can approximate the function $\phi_{d_i}(V_i V_i^\top, \lambda_i) = \lambda_i$ and $\rho \approx \max_{i=1}^{l}$.

This in fact gives an infinite family of graphs that SignNet / BasisNet can distinguish, but spectral graph convolutions or spectral graph GNNs cannot. To see why, suppose we have $G^{(1)}$ and $G^{(2)}$ for some $n \geq 5$. Then we construct a pair of graphs on $n + 1$ nodes with the same degree sequence. To do this, we add another node to the path of $G^{(1)}$, thus giving it degree sequence $\{3, 2, \ldots, 2, 1\}$. For $G^{(2)}$, we add a node $v_{n+1}$ to the side that $v_n$ is not contained on (e.g. for $n = 5$, we add $v_6$ to the left side, as $v_5$ was on the right), then connect $v_n$ to $v_{n+1}$ to also give a degree sequence $\{3, 2, \ldots, 2, 1\}$. Note that the non-bipartiteness of $G^{(1)}$ and bipartiteness of $G^{(2)}$ are preserved.

$\square$

## H.2 Existing Positional Encodings

Here, we show that our SignNets and BasisNets universally approximate various types of existing graph positional encodings. The key is to show that these positional encodings are related to spectral graph convolution matrices and the diagonals of these matrices, and to show that our networks can approximate these matrices and diagonals.

**Proposition 5.** *If the eigenvalues take values in a compact set, SignNets and BasisNets universally approximate the diagonal of any spectral graph convolution matrix $f(V, \Lambda) = \text{diag}\left(\sum_{i=1}^{n} h(\lambda_i) v_i v_i^\top\right)$. BasisNets can additionally universally approximate any spectral graph convolution matrix $f(V, \Lambda) = \sum_{i=1}^{n} h(\lambda_i) v_i v_i^\top$.*

*Proof.* Note that the $v_i$ come from a compact set as they are of unit norm. The $\lambda_i$ are from a compact set by assumption; this assumption holds for the normalized Laplacian, as $\lambda_i \in [0, 2]$. Also, as $\text{diag}$ is linear, the spectral graph convolution diagonal can be written $\sum_{i=1}^{n} h(\lambda_i) \text{diag}(v_i v_i^\top)$.

Let $\epsilon > 0$. For SignNet, let $\rho = \sum_{i=1}^{n}$, which can be exactly expressed as it is a permutation equivariant linear operation from vectors to vectors. Then $\phi(v_i, \lambda_i)$ can approximate the function $\lambda_i \text{diag}(v_i v_i^\top)$ to arbitrary precision, as it is a permutation equivariant function from vectors to vectors [Segol and Lipman, 2019]. Thus, letting $\phi$ approximate the function to $\epsilon/n$ accuracy, SignNet can approximate $f$ to $\epsilon$ accuracy.

Let $l$ be the number of eigenspaces $V_1, \ldots, V_l$, so $f(V, \Lambda) = \sum_{i=1}^{l} h(\mu_i) V_i V_i^\top$. For BasisNet, we need only show that it can approximate the spectral graph convolution matrix to $\epsilon/l$ accuracy, as a 2-IGN can exactly express the $\text{diag}$ function in each $\phi_{d_i}$, since it is a linear permutation equivariant function from matrices to vectors. A 2-IGN can universally approximate the function $f_1(\mu_i, V_i V_i^\top) = (h(\mu_i), V_i V_i^\top)$, as it can express any elementwise MLP. Also, a 2-IGN can universally approximate the scalar-matrix multiplication $f_2(h(\mu_i), V_i V_i^\top) = h(\mu_i) V_i V_i^\top$ by another elementwise MLP. Since $h(\mu_i) V_i V_i^\top = f_2 \circ f_1(\mu_i, V_i V_i^\top)$, Lemma 6 shows that a single 2-IGN can approximate this composition to $\epsilon/l$ accuracy, so we are done.

$\qquad\square$

**Proposition 4.** *SignNet and BasisNet universally approximate node positional encodings based on heat kernels [Feldman et al., 2022] and random walks [Dwivedi et al., 2022]. BasisNet universally approximates diffusion and $p$-step random walk relative positional encodings [Mialon et al., 2021], and generalized PageRank and landing probability distance encodings [Li et al., 2020].*

*Proof.* We will show that we can apply the above Proposition 5, by showing that all of these positional encodings are spectral graph convolutions. The heat kernel embeddings are of the form $\text{diag}\left(\sum_{i=1}^{n} \exp(-t\lambda_i) v_i v_i^\top\right)$ for some choices of the parameter $t$, so they can be approximated by SignNets or BasisNets. Also, the diffusion kernel [Mialon et al., 2021] is just the matrix of this heat kernel, and the $p$-step random walk kernel is $\sum_{i=1}^{n}(1 - \gamma\lambda_i)^p v_i v_i^\top$ for some parameter $\gamma$, so BasisNets can universally approximate both of these.

For the other positional encodings, we let $v_i$ be the eigenvectors of the random walk Laplacian $I - D^{-1}A$ instead of the normalized Laplacian $I - D^{-1/2}AD^{-1/2}$. The eigenvalues of these two Laplacians are the same, and if $\tilde{v}_i$ is an eigenvector of the normalized Laplacian then $D^{-1/2}\tilde{v}_i$ is an eigenvector of the random walk Laplacian with the same eigenvalue [Von Luxburg, 2007].

Then with $v_i$ as the eigenvectors of the random walk Laplacian, the random walk positional encodings (RWPE) in Dwivedi et al. [2022] take the form

$$\text{diag}\left((D^{-1}A)^k\right) = \text{diag}\left(\sum_{i=1}^{n}(1 - \lambda_i)^k v_i v_i^\top\right), \qquad (45)$$

for any choices of integer $k$.

The distance encodings proposed in Li et al. [2020] take the form

$$f_3(AD^{-1}, (AD^{-1})^2, (AD^{-1})^3, \ldots), \qquad (46)$$

for some function $f_3$. We restrict to continuous $f_3$ here; shortest path distances can be obtained by a discontinuous $f_3$ that we discuss below. Their generalized PageRank based distance encodings can be obtained by

$$\sum_{i=1}^{n}\left(\sum_{k \geq 1} \gamma_k (1 - \lambda_i)^k\right) v_i v_i^\top \qquad (47)$$

for some $\gamma_k \in \mathbb{R}$, so this is a spectral graph convolution. They also define so-called landing probability based positional encodings, which take the form

$$\sum_{i=1}^{n}(1 - \lambda_i)^k v_i v_i^\top, \qquad (48)$$

for some choices of integer $k$. Thus, BasisNets can approximate these distance encoding matrices. $\quad\square$

Another powerful class of positional encodings is based on shortest path distances between nodes in the graph [Ying et al., 2021, Li et al., 2020]. Shortest path distances can be expressed in a

form similar to the spectral graph convolution, but require a highly discontinuous function. If we define $f_3(x_1, \ldots, x_n) = \min_{i:x_i \neq 0} i$ to be the lowest index such that $x_i$ is nonzero, then we can write the shortest path distance matrix as $f_3(D^{-1}A, (D^{-1}A)^2, \ldots, (D^{-1}A)^n)$, where $f_3$ is applied elementwise to return an $n \times n$ matrix. As $(D^{-1}A)^k = \sum_{i=1}^{n}(1-\lambda_i)^k v_i v_i^\top$, BasisNets can learn the inside arguments, but cannot learn the discontinuous function $f_3$.

## H.3 Spectral Invariants

Here, we consider the graph angles $\alpha_{ij} = \|V_i V_i^\top e_j\|_2$, for $i = 1, \ldots, l$ where $l$ is the number of eigenspaces, and $j = 1, \ldots, n$. It is clear that graph angles are permutation equivariant and basis invariant. These graph angles have been extensively studied, so we cite a number of interesting properties of them. That graph angles determine the number of length 3, 4 and 5 cycles, the connectivity of a graph, and the number of length $k$ closed walks is all shown in Chapter 4 of Cvetković et al. [1997]. Other properties may be of use for graph representation learning as well. For instance, the eigenvalues of node-deleted subgraphs of a graph $\mathcal{G}$ are determined by the eigenvalues and graph angles of $\mathcal{G}$; this may be useful in extending recent graph neural networks that are motivated by node deletion and the reconstruction conjecture [Cotta et al., 2021, Bevilacqua et al., 2022, Papp et al., 2021, Tahmasebi et al., 2020].

Now, we prove that BasisNet can universally approximate the graph angles. The graph properties we consider in the theorem are all integer valued (e.g. the number of cycles of length 3 in a graph is an integer). Thus, any two graphs that differ in these properties will differ by at least 1, so as long as we have approximation to $\varepsilon < 1/2$, we can distinguish any two graphs that differ in these properties. Recall the statement of Theorem 2.

**Theorem 2.** *BasisNet can universally approximate the graph angles $\alpha_{ij}$. The eigenvalues and graph angles (and thus BasisNets) can determine the number of length 3, 4, and 5 cycles, whether a graph is connected, and the number of length $k$ closed walks from any vertex to itself.*

*Proof.* Note that the graph angles satisfy

$$\alpha_{ij} = \|V_i V_i^\top e_j\|_2 = \sqrt{e_j^\top V_i V_i^\top V_i V_i^\top e_j} = \sqrt{e_j^\top V_i V_i^\top e_j}, \tag{49}$$

where $V_i$ is a basis for the $i$th adjacency matrix eigenspace, and $e_j^\top V_i V_i^\top e_j$ is the $(j,j)$-entry of $V_i V_i^\top$. These graph angles are just the elementwise square roots of the diagonals of the matrices $V_i V_i^\top$. As $f_1(V_i V_i^\top) = \mathrm{diag}(V_i V_i^\top)$ is a permutation equivariant linear function from matrices to vectors, 2-IGN on $V_i V_i^\top$ can exactly compute this with 0 error. Then a 2-IGN can learn an elementwise MLP to approximate the elementwise square root $f_2(\mathrm{diag}(V_i V_i^\top)) = \sqrt{\mathrm{diag}(V_i V_i^\top)}$ to arbitrary precision. Finally, there may be remaining operations $f_3$ that are permutation invariant or permutation equivariant from vectors to vectors; for instance, the $\alpha_{ij}$ are typically gathered into a matrix of size $l \times n$ where the columns are lexicographically sorted ($l$ is the number of eigenspaces) [Cvetković et al., 1997], or we may have a permutation invariant readout to compute a subgraph count. A DeepSets can approximate $f_3$ without any higher order tensors besides vectors [Zaheer et al., 2017, Segol and Lipman, 2019].

As 2-IGNs can approximate each $f_i$ individually, a single 2-IGN can approximate $f_3 \circ f_2 \circ f_1$ by Lemma 6. Also, since the graph properties considered in the theorem are integer-valued, BasisNet can distinguish any two graphs that differ in one of these properties. $\square$

To see that message passing graph neural networks (MPNNs) cannot determine these quantities, we use the fact that MPNNs cannot distinguish between two graphs that have the same number of nodes and where each node (in both graphs) has the same degree. For $k \geq 3$, let $C_k$ denote the cycle graph of size $k$, and $C_k + C_k$ denote the graph that is the union of two disjoint cycle graphs of size $k$. MPNNs cannot distinguish between $C_{2k}$ and $C_k + C_k$ for $k \geq 3$, because they have the same number of nodes, and each node has degree 2. Thus, MPNNs cannot tell whether a graph is connected, as $C_{2k}$ is but $C_k + C_k$ is not. Also, it cannot count the number of 3, 4, or 5 cycles, as $C_k + C_k$ has two $k$ cycles while $C_{2k}$ has no $k$ cycles. Likewise, any node in $C_k + C_k$ has more length $k$ closed walks than any node in $C_{2k}$. This is because any length $k$ closed walk in $C_{2k}$ has an analogous closed walk in $C_k + C_k$, but the nodes in $C_k + C_k$ also have a closed walk that completely goes around a cycle.

# I Useful Lemmas

In this section, we collect useful lemmas for our proofs. These lemmas generally only require basic tools to prove. Our first lemma is a crucial property of quotient spaces.

**Lemma 1** (Passing to the quotient). *Let $\mathcal{X}$ and $\mathcal{Y}$ be topological spaces, and let $\mathcal{X}/G$ be a quotient space, with corresponding quotient map $\pi$. Then for every continuous $G$-invariant function $f : \mathcal{X} \to \mathcal{Y}$, there is a unique continuous $\tilde{f} : \mathcal{X}/G \to \mathcal{Y}$ such that $f = \tilde{f} \circ \pi$.*

*Proof.* For $z \in \mathcal{X}/G$, by surjectivity of $\pi$ we can choose an $x_z \in \mathcal{X}$ such that $\pi(x_z) = z$. Define $\tilde{f} : \mathcal{X}/G \to \mathcal{Y}$ by $\tilde{f}(z) = f(x_z)$. This is well-defined, since if $\pi(x_z) = \pi(x)$ for any other $x \in \mathcal{X}$, then $gx_z = x$ for some $g \in G$, so

$$f(x) = f(gx_z) = f(x_z) = \tilde{f}(z), \tag{50}$$

where the second equality uses the $G$-invariance of $f$. Note that $\tilde{f}$ is continuous by the universal property of quotient spaces. Also, $\tilde{f}$ is the unique function such that $f = \tilde{f} \circ \pi$; if there were another function $h : \mathcal{X}/G \to \mathcal{Y}$ with $h(z) \neq \tilde{f}(z)$, then $h(z) \neq f(x_z)$, so $h(\pi(x_z)) = h(z) \neq f(x_z)$. $\square$

Next, we give the First Fundamental Theorem of $O(d)$, a classical result that has been recently used for machine learning by Villar et al. [2021]. This result shows that an orthogonally invariant $f(V)$ can be expressed as a function $h(VV^\top)$. We give a proof that if $f$ is continuous, then $h$ is also continuous.

**Lemma 2** (First Fundamental Theorem of $O(d)$). *A continuous function $f : \mathbb{R}^{n \times d} \to \mathbb{R}^s$ is orthogonally invariant, i.e. $f(VQ) = f(V)$ for all $Q \in O(d)$, if and only if $f(V) = h(VV^\top)$ for some continuous $h$.*

*Proof.* If $f(V) = h(VV^\top)$, then we have $f(VQ) = h(VQQ^\top V^\top) = h(VV^\top)$ so $f$ is orthogonally invariant.

For the other direction, invariant theory shows that the $O(d)$ invariant polynomials are generated by the inner products $v_i^\top v_j$, where $v_i \in \mathbb{R}^d$ are the rows of $V$ [Kraft and Procesi, 1996]. Let $p : \mathbb{R}^{n \times d} \to \mathbb{R}^{n \times n}$ be the map $p(V) = VV^\top$. Then González and de Salas [2003] Lemma 11.13 shows that the quotient space $\mathbb{R}^{n \times d}/O(d)$ is homeomorphic to a closed subset $p(\mathbb{R}^{n \times d}) = \mathcal{Z} \subseteq \mathbb{R}^{n \times n}$. Let $\tilde{p}$ refer to this homeomorphism, and note that $\tilde{p} \circ \pi = p$ by passing to the quotient (Lemma 1). Then any continuous $O(d)$ invariant $f$ passes to a unique continuous $\tilde{f} : \mathbb{R}^{n \times d}/O(d) \to \mathbb{R}^s$ (Lemma 1), so $f = \tilde{f} \circ \pi$ where $\pi$ is the quotient map. Define $h : \mathcal{Z} \to \mathbb{R}^s$ by $h = \tilde{f} \circ \tilde{p}^{-1}$, and note that $h$ is a composition of continuous functions and hence continuous. Finally, we have that $h(VV^\top) = h(\tilde{p} \circ \pi(V)) = \tilde{f} \circ \pi(V) = f(V)$, so we are done. $\square$

The next lemma allows us to decompose a quotient of a product space into a product of smaller quotient spaces.

**Lemma 3.** *Let $\mathcal{X}_1, \ldots, \mathcal{X}_k$ be topological spaces and $G_1, \ldots, G_k$ be topological groups such that each $G_i$ acts continuously on $\mathcal{X}_i$. Denote the quotient maps by $\pi_i : \mathcal{X}_i \to \mathcal{X}_i/G_i$. Then the quotient of the product is the product of the quotient, i.e.*

$$(\mathcal{X}_1 \times \ldots \times \mathcal{X}_k)/(G_1 \times \ldots \times G_k) \cong (\mathcal{X}_1/G_1) \times \ldots \times (\mathcal{X}_k/G_k), \tag{51}$$

*and $\pi_1 \times \ldots \times \pi_k : \mathcal{X}_1 \times \ldots \mathcal{X}_k \to (\mathcal{X}_1/G_1) \times \ldots \times (\mathcal{X}_k/G_k)$ is quotient map.*

*Proof.* First, we show that $\pi_1 \times \ldots \times \pi_k$ is a quotient map. This is because 1. the quotient map of any continuous group action is an open map, so each $\pi_i$ is an open map, 2. the product of open maps is an open map, so $\pi_1 \times \ldots \times \pi_k$ is an open map and 3. a continuous surjective open map is a quotient map, so $\pi_1 \times \ldots \times \pi_k$, which is continuous and surjective, is a quotient map.

Now, we need only apply the theorem of uniqueness of quotient spaces to show (51) (see e.g. Lee [2013], Theorem A.31). Letting $q : \mathcal{X}_1 \times \ldots \times \mathcal{X}_k \to (\mathcal{X}_1 \times \ldots \times \mathcal{X}_k)/(G_1 \times \ldots \times G_k)$ denote the quotient map for this space, it is easily seen that $q(x_1, \ldots, x_k) = q(y_1 \ldots, y_k)$ if and only if $\pi_1 \times \ldots \times \pi_k(x_1, \ldots, x_k) = \pi_1 \times \ldots \times \pi_k(y_1, \ldots, y_k)$, since either of these is true if and only if there exist $g_i \in G_i$ such that $x_i = g_i y_i$ for each $i$. Thus, we have an isomorphism of these quotient spaces. $\square$

The following lemma shows that quotients of compact spaces are also compact, which is useful for universal approximation on quotient spaces.

**Lemma 4** (Compactness of quotients of compact spaces). *Let $\mathcal{X}$ be a compact space. Then the quotient space $\mathcal{X}/G$ is compact.*

*Proof.* Denoting the quotient map by $\pi : \mathcal{X} \to \mathcal{X}/G$ and letting $\{U_\alpha\}_\alpha$ be an open cover of $\mathcal{X}/G$, we have that $\{\pi^{-1}(U_\alpha)\}_\alpha$ is an open cover of $\mathcal{X}$. By compactness of $\mathcal{X}$, we can choose a finite subcover $\{\pi^{-1}(U_{\alpha_i})\}_{i=1,\dots,n}$. Then $\{\pi(\pi^{-1}(U_{\alpha_i}))\}_{i=1,\dots,n} = \{U_{\alpha_i}\}_{i=1,\dots,n}$ by surjectivity, and $\{U_{\alpha_i}\}_{i=1,\dots,n}$ is thus an open cover of $\mathcal{X}/G$. $\qquad\square$

The Whitney embedding theorem gives a nice condition that we apply to show that the quotient spaces $\mathcal{X}/G$ that we deal with embed into Euclidean space. It says that when $\mathcal{X}/G$ is a smooth manifold, then it can be embedded into a Euclidean space of double the dimension of the manifold. The proof is outside the scope of this paper.

**Lemma 5** (Whitney Embedding Theorem [Whitney, 1944]). *Every smooth manifold $\mathcal{M}$ of dimension $n > 0$ can be smoothly embedded in $\mathbb{R}^{2n}$.*

Finally, we give a lemma that helps prove universal approximation results. It says that if functions $f$ that we want to approximate can be written as compositions $f = f_L \circ \dots \circ f_1$, then it suffices to universally approximate each $f_i$ and compose the results to universally approximate the $f$. This is especially useful for proving universality of neural networks, as we may use some layers to approximate each $f_i$, then compose these layers to approximate the target function $f$.

**Lemma 6** (Layer-wise universality implies universality). *Let $\mathcal{Z} \subseteq \mathbb{R}^{d_0}$ be a compact domain, let $\mathcal{F}_1, \dots, \mathcal{F}_L$ be families of continuous functions where $\mathcal{F}_i$ consists of functions from $\mathbb{R}^{d_{i-1}} \to \mathbb{R}^{d_i}$ for some $d_1, \dots, d_L$. Let $\mathcal{F}$ be the family of functions $\{f_L \circ \dots f_1 : \mathcal{Z} \to \mathbb{R}^{d_L}, f_i \in \mathcal{F}_i\}$ that are compositions of functions $f_i \in \mathcal{F}_i$.*

*For each $i$, let $\Phi_i$ be a family of continuous functions that universally approximates $\mathcal{F}_i$. Then the family of compositions $\Phi = \{\phi_L \circ \dots \circ \phi_1 : \phi_i \in \Phi_i\}$ universally approximates $\mathcal{F}$.*

*Proof.* Let $f = f_L \circ \dots \circ f_1 \in \mathcal{F}$. Let $\tilde{\mathcal{Z}}_1 = \mathcal{Z}$, and then for $i \geq 2$ let $\tilde{\mathcal{Z}}_i = f_{i-1}(\tilde{\mathcal{Z}}_{i-1})$. Then each $\tilde{\mathcal{Z}}_i$ is compact by continuity of the $f_i$. For $1 \leq i < L$, let $\mathcal{Z}_i = \tilde{\mathcal{Z}}_i$, and for $i = L$ let $\mathcal{Z}_L$ be a compact set containing $\tilde{\mathcal{Z}}_L$ such that every ball of radius one centered at a point in $\tilde{\mathcal{Z}}_L$ is still contained in $\mathcal{Z}_L$.

Let $\epsilon > 0$. We will show that there is a $\phi \in \Phi$ such that $\|f - \phi\|_\infty < \epsilon$ by induction on $L$. This holds trivially for $L = 1$, as then $\Phi = \Phi_1$.

Now, let $L \geq 2$, and suppose it holds for $L - 1$. By universality of $\Phi_L$, we can choose a $\phi_L : \mathcal{Z}_L \to \mathbb{R}^{d_L} \in \Phi_L$ such that $\|\phi_L - f_L\|_\infty < \epsilon/2$. As $\phi_L$ is continuous on a compact domain, it is also uniformly continuous, so we can choose a $\tilde{\delta} > 0$ such that $\|y - z\|_2 < \tilde{\delta} \implies \|\phi_L(y) - \phi_L(z)\|_2 < \epsilon/2$.

Let $\delta = \min(\tilde{\delta}, 1)$. By induction, we can choose $\phi_{L-1} \circ \dots \circ \phi_1, \phi_i \in \Phi_i$ such that

$$\|\phi_{L-1} \circ \dots \circ \phi_1 - f_{L-1} \circ \dots \circ f_1\|_\infty < \delta. \tag{52}$$

Note that $\phi_{L-1} \circ \dots \circ \phi_1(\mathcal{Z}) \subseteq \mathcal{Z}_L$, because for each $x \in \mathcal{Z}$, $\phi_{L-1} \circ \dots \circ \phi_1(x)$ is within $\delta \leq 1$ Euclidean distance to $f_{L-1} \circ \dots \circ f_1(x) \in \tilde{\mathcal{Z}}_L$, so it is contained in $\mathcal{Z}_L$ by construction. Thus, we may define $\phi = \phi_L \circ \dots \circ \phi_1 : \mathcal{Z} \to \mathbb{R}^{d_L}$, and compute that

$$\|\phi - f\|_\infty \leq \|\phi - \phi_L \circ f_{L-1} \circ \dots \circ f_1\|_\infty + \|\phi_L \circ f_{L-1} \circ \dots \circ f_1 - f\|_\infty \tag{53}$$
$$< \|\phi - \phi_L \circ f_{L-1} \circ \dots \circ f_1\|_\infty + \epsilon/2, \tag{54}$$

since $\|\phi_L - f_L\|_\infty < \epsilon/2$. To bound this other term, let $x \in \mathcal{Z}$, and for $y = \phi_{L-1} \circ \dots \circ \phi_1(x)$ and $z = f_{L-1} \circ \dots \circ f_1(x)$, we know that $\|y - z\|_2 < \delta$, so $\|\phi_L(y) - \phi_L(z)\|_2 < \epsilon/2$ by uniform continuity. As this holds for all $x$, we have $\|\phi - \phi_L \circ f_{L-1} \circ \dots \circ f_1\|_\infty \leq \epsilon/2$, so $\|\phi - f\|_\infty < \epsilon$ and we are done. $\qquad\square$

Table 7: Results on the ZINC dataset with 500k parameter budget and no edge features. Numbers are the mean and standard deviation over 4 runs each with different seeds.

| Base model | Positional encoding | $k$ | #params | Test MAE ($\downarrow$) |
|---|---|---|---|---|
| | No PE | 16 | 497k | $0.348_{\pm 0.014}$ |
| GIN | LapPE (flip) | 16 | 498k | $0.341_{\pm 0.011}$ |
| | SignNet | 16 | 500k | $\mathbf{0.238}_{\pm \mathbf{0.012}}$ |
| | No PE | 16 | 501k | $0.464_{\pm 0.011}$ |
| GAT | LapPE (flip) | 16 | 502k | $0.462_{\pm 0.013}$ |
| | SignNet | 16 | 499k | $\mathbf{0.243}_{\pm \mathbf{0.008}}$ |

## J   Further Experiments

### J.1   Graph Regression with no Edge Features

All graph regression models in Table 1 use edge features for learning and inference. To show that SignNet is also useful when no edge features are available, we ran ZINC experiments without edge features as well. The results are displayed in Table 7. In this setting, SignNet still significantly improves the performance over message passing networks without positional encodings, and over Laplacian positional encodings with sign flipping data augmentation.

### J.2   Learning Spectral Graph Convolutions

Table 8: Sum of squared errors for spectral graph convolution regression (with no test set). Lower is better. Numbers are mean and standard deviation over 50 images from He et al. [2021].

| | Low-pass | High-pass | Band-pass | Band-rejection | Comb |
|---|---|---|---|---|---|
| GCN | $.111_{\pm .068}$ | $3.092_{\pm 5.11}$ | $1.720_{\pm 3.15}$ | $1.418_{\pm 1.03}$ | $1.753_{\pm 1.17}$ |
| GAT | $.113_{\pm .065}$ | $.954_{\pm .696}$ | $1.105_{\pm .964}$ | $.543_{\pm .340}$ | $.638_{\pm .446}$ |
| GPR-GNN | $.033_{\pm .032}$ | $.012_{\pm .007}$ | $.137_{\pm .081}$ | $.256_{\pm .197}$ | $.369_{\pm .460}$ |
| ARMA | $.053_{\pm .029}$ | $.042_{\pm .024}$ | $.107_{\pm .039}$ | $.148_{\pm .089}$ | $.202_{\pm .116}$ |
| ChebNet | $.003_{\pm .002}$ | $\mathbf{.001}_{\pm .001}$ | $.005_{\pm .003}$ | $.009_{\pm .006}$ | $.022_{\pm .016}$ |
| BernNet | $\mathbf{.001}_{\pm .002}$ | $\mathbf{.001}_{\pm .001}$ | $\mathbf{.000}_{\pm .000}$ | $.048_{\pm .042}$ | $.027_{\pm .019}$ |
| Transformer | $3.662_{\pm 1.97}$ | $3.715_{\pm 1.98}$ | $1.531_{\pm 1.30}$ | $1.506_{\pm 1.29}$ | $3.178_{\pm 1.93}$ |
| Transformer Eig Flip | $4.454_{\pm 2.32}$ | $4.425_{\pm 2.38}$ | $1.651_{\pm 1.53}$ | $2.567_{\pm 1.73}$ | $3.720_{\pm 1.94}$ |
| Transformer Eig Abs | $2.727_{\pm 1.40}$ | $3.172_{\pm 1.61}$ | $1.264_{\pm .788}$ | $1.445_{\pm .943}$ | $2.607_{\pm 1.32}$ |
| DeepSets SignNet | $.004_{\pm .013}$ | $.086_{\pm .405}$ | $.021_{\pm .115}$ | $.008_{\pm .037}$ | $\mathbf{.003}_{\pm .016}$ |
| Transformer SignNet | $.003_{\pm .016}$ | $.004_{\pm .025}$ | $.001_{\pm .004}$ | $.006_{\pm .023}$ | $.093_{\pm .641}$ |
| DeepSets BasisNet | $.009_{\pm .018}$ | $.003_{\pm .015}$ | $.008_{\pm .030}$ | $\mathbf{.004}_{\pm .011}$ | $.015_{\pm .060}$ |
| Transformer BasisNet | $.079_{\pm .471}$ | $.014_{\pm .038}$ | $.005_{\pm .018}$ | $.006_{\pm .016}$ | $.014_{\pm .051}$ |

To numerically test the ability of our basis invariant networks for learning spectral graph convolutions, we follow the experimental setups of Balcilar et al. [2020], He et al. [2021]. We take the dataset of 50 images in He et al. [2021] (originally from the Image Processing Toolbox of MATLAB), and resize them from $100{\times}100$ to $32{\times}32$. Then we apply the same spectral graph convolutions on them as in He et al. [2021], and train neural networks to learn these as regression targets. As in prior work, we report sum of squared errors on the training set to measure expressivity.

We compare against message passing GNNs [Kipf and Welling, 2017, Veličković et al., 2018] and spectral GNNs [Chien et al., 2021, Bianchi et al., 2021, Defferrard et al., 2016, He et al., 2021]. Also, we consider standard Transformers with only node features, with eigenvectors and sign flip augmentation, and with absolute values of eigenvectors. These models are all approximately sign invariant (they either use eigenvectors in a sign invariant way or do not use eigenvectors). We use DeepSets [Zaheer et al., 2017] in SignNet and 2-IGN [Maron et al., 2018] in BasisNet for $\phi$, use a DeepSets for $\rho$ in both cases, and then feed the features into another DeepSets or a standard Transformer [Vaswani et al., 2017] to make the final predictions. That is, we are only given graph information through the eigenvectors and eigenvalues, and we do not use message passing.

Table 8 displays the results, which validate our theoretical results in Section 3.1. Without any message passing, SignNet and BasisNet allow DeepSets and Transformers to perform strongly, beating the spectral GNNs GPR-GNN and ARMA on all tasks. Also, our networks outperform all other methods on the band-rejection and comb filters, and are mostly close to the best model on the other filters.

## K  Further Experimental Details

### K.1  Hardware, Software, and Data Details

All experiments could fit on one GPU at a time. Most experiments were run on a server with 8 NVIDIA RTX 2080 Ti GPUs. We run all of our experiments in Python, using the PyTorch [Paszke et al., 2019] framework (license URL). We also make use of Deep Graph Library (DGL) [Wang et al., 2019] (Apache License 2.0), and PyTorch Geometric (PyG) [Fey and Lenssen, 2019] (MIT License) for experiments with graph data.

We open source our code [redacted for anonymous review].

The data we use are all freely available online. The datasets we use are ZINC [Irwin et al., 2012], Alchemy [Chen et al., 2019a], the synthetic counting substructures dataset [Chen et al., 2020], the multi-task graph property regression synthetic dataset [Corso et al., 2020] (MIT License), the images dataset used by Balcilar et al. [2020] (GNU General Public License v3.0), the cat mesh from free3d.com/3d-model/cat-v1--522281.html (Personal Use License), and the human mesh from turbosquid.com/3d-models/water-park-slides-3d-max/1093267 (TurboSquid 3D Model License). If no license is listed, this means that we cannot find a license for the dataset. As they appear to be freely available with permissive licenses or no licenses, we do not ask for permission from the creators or hosts of the data.

We do not believe that any of this data contains offensive content or personally identifiable information. The 50 images used in the spectral graph convolution experiments are mostly images of objects, with a few low resolution images of humans that do not appear to have offensive content. The only other human-related data appears to be the human mesh, which appears to be from a 3D scan of a human. The human mesh does have tattoos, but they do not appear to be offensive.

### K.2  Graph Regression Details

**ZINC.** In Section 4.1 we study the effectiveness of SignNet for learning positional encodings to boost the expressive power, and thereby generalization, on the graph regression problem ZINC. In all cases we take our $\phi$ encoder to be an 8 layer GIN with ReLU activation. The input eigenvector $v_i \in \mathbb{R}^n$, where $n$ is the number of nodes in the graph, is treated as a single scalar feature for each node. In the case of using a fixed number of eigenvectors $k$, the aggregator $\rho$ is taken to be an 8 layer MLP with batch normalization and ReLU activation. The aggregator $\rho$ is applied separately to the concatenatation of the $k$ different embeddings for each node in a graph, resulting in one single embedding per node. This embedding is concatenated to the node features for that node, and the result passed as input to the base (predictor) model. We also consider using all available eigenvectors in each graph instead of a fixed number $k$. Since the total number of eigenvectors is a variable quantity, equal to the number of nodes in the underlying graph, an MLP cannot be used for $\rho$. To handle the variable sized input in this case, we take $\rho$ to be an MLP preceded by a sum over the $\phi$ outputs. In other words, the SignNet is of the form $\text{MLP}\left(\sum_{i=1}^{k} \phi(v_i) + \phi(-v_i)\right)$ in this case.

As well as testing SignNet, we also checked whether simple transformations that resolve the sign ambiguity of the Laplacian eigenvectors $p = (v_1, \ldots, v_k)$ could serve as effective positional encoding. We considered three options. First is to randomly flip the sign of each $\pm v_i$ during training. This is a common heuristic used in prior work on Laplacian positional encoding [Kreuzer et al., 2021, Dwivedi et al., 2020]. Second, take the element-wise absolute value $|v_i|$. This is a non-injective map, creating sign invariance at the cost of destroying positional information. Third is a different canonicalization that avoids stochasticity and use of absolute values by selecting the sign of each $v_i$ so that the majority of entries are non-negative, with ties broken by comparing the $\ell_1$-norm of positive and negative parts. When the tie-break also fails, the sign is chosen randomly. Results for GatedGCN base model on ZINC in Table 1 show that all three of these approaches are significantly poorer positional encodings compared to SignNet.

Our training pipeline largely follows that of Dwivedi et al. [2022], and we use the GatedGCN and PNA base models from the accompanying implementation (see https://github.com/vijaydwivedi75/gnn-lspe). The Sparse Transformer base model architecture we use, which like GAT computes attention only across neighbouring nodes, is introduced by Kreuzer et al. [2021]. Finally, the GINE implementation is based on the PyTorch Geometric implementation [Fey and Lenssen, 2019]. For the state-of-the-art comparison, all baseline results are from their respective papers, except for GIN, which we run.

**ZINC-full.** We also run our method on the full ZINC dataset, termed ZINC-full. The result we report for SignNet is a larger version of the GatedGCN base model with a SignNet that takes in all eigenvectors. This model has 994,113 parameters in total. All baseline results are from their respective papers, except for GIN, which is from [Bodnar et al., 2021].

**Alchemy.** We run our method and compare with the state-of-the-art on Alchemy (with 10,000 training graphs). We use the same data split as Morris et al. [2020b]. Our base model is a GIN that takes in edge features (i.e. a GINE). The SignNet consists of GIN for $\phi$ and a Transformer for $\rho$, as in the counting substructures and graph property regression experiments in Section 4.2. The model has 907,371 parameters in total. Our training setting is very similar to that of Morris et al. [2022], as we build off of their code. We train with an Adam optimizer [Kingma and Ba, 2014] with a starting learning rate of .001, and a minimum learning rate of .000001. The learning rate schedule cuts the learning rate in half with a patience of 20 epochs, and training ends when we reach the minimum learning rate. All baseline results are from their respective papers, except for GIN, which is from [Morris et al., 2022].

### K.3 Spectral Graph Convolution Details

In Appendix J.2, we conduct node regression experiments for learning spectral graph convolutions. The experimental setup is mostly taken from He et al. [2021]. However, we resize the $100 \times 100$ images to $32 \times 32$. Thus, each image is viewed as a 1024-node graph. The node features $X \in \mathbb{R}^n$ are the grayscale pixel intensities of each node. Just as in He et al. [2021], we only train and evaluate on nodes that are not connected to the boundary of the grid (that is, we only evaluate on the $28 \times 28$ middle section). For all experiments we limit each model to 50,000 parameters. We use the Adam [Kingma and Ba, 2014] optimizer for all experiments. For each of the GNN baselines (GCN, GAT, GPR-GNN, ARMA, ChebNet, BernNet), we select the best performing out of 4 hyperparameter settings: either 2 or 4 convolution layers, and a hidden dimension of size 32 or $D$, where $D$ is just large enough to stay with 50,000 parameters (for instance, $D = 128$ for GCN, GPR-GNN, and BernNet).

We use DeepSets or standard Transformers as our prediction network. This takes in the output of SignNet or BasisNet and concatenates it with the node features, then outputs a scalar prediction for each node. We use a 3 layer output network for DeepSets SignNet, and 2 layer output networks for all other configurations. All networks use ReLU activations.

For SignNet, we use DeepSets for both $\phi$ and $\rho$. Our $\phi$ takes in eigenvectors only, then our $\rho$ takes the outputs of $\phi$ and the eigenvalues. We use three layers for $\phi$ and $\rho$.

For BasisNet, we use the same DeepSets for $\rho$ as in SignNet, and 2-IGNs for the $\phi_{d_i}$. There are three distinct multiplicities for the grid graph (1, 2, and 32), so we only need 3 separate IGNs. Each IGN consists of an $\mathbb{R}^{n^2 \times 1} \to \mathbb{R}^{n \times d'}$ layer and two $\mathbb{R}^{n \times d''} \to \mathbb{R}^{n \times d'''}$ layers, where the $d'$ are hidden dimensions. There are no matrix to matrix operations used, as the memory requirements are intensive for these $\geq 1000$ node graphs. The $\phi_{d_i}$ only take in $V_i V_i^\top$ from the eigenspaces, and the $\rho$ takes the output of the $\phi_{d_i}$ as well as the eigenvalues.

### K.4 Substructures and Graph Properties Regression Details

We use the random graph dataset from Chen et al. [2020] for counting substructures and the synthetic dataset from Corso et al. [2020] for regressing graph properties. For fair comparison we fix the base model as a 4-layer GIN model with hidden size 128. We choose $\phi$ as 4-layer GIN (independently applied to every eigenvector) and $\rho$ as 1-layer Transformer (independently applied to every node). Combined with proper batching and masking, we have a SignNet that takes Laplacian eigenvectors $V \in \mathbb{R}^{n \times n}$ and outputs fixed size sign-invariant encoding node features $f(V, \Lambda, X) \in \mathbb{R}^{n \times d}$, where

1574   $n$ varies between graphs but $d$ is fixed. We use this SignNet in our experiments and compare with
1575   other methods of handling PEs.

## K.5   Texture Reconstruction Details

Table 9: Parameter settings for the texture reconstruction experiments.

|  | Params | Base MLP width | Base MLP layers | $\phi$ out dim | $\rho$ out dim | $\rho$, $\phi$ width |
|---|---|---|---|---|---|---|
| Intrinsic NF | 328,579 | 128 | 6 | — | — | — |
| SignNet | 323,563 | 108 | 6 | 4 | 64 | 8 |

1577   We closely follow the experimental setting of Koestler et al. [2022] for the texture reconstruction
1578   experiments. In this work, we use the cotangent Laplacian [Rustamov et al., 2007] of a triangle mesh
1579   with the lowest 1023 eigenvectors besides the trivial eigenvector of eigenvalue 0. We implemented
1580   SignNet in the authors' original code, which was privately shared with us. Both $\rho$ and $\phi$ are taken
1581   to be MLPs. Hyperparameter settings and number of parameters are given in Table 9. We chose
1582   hyperparameters so that the total number of parameters in the SignNet model was no larger than that
1583   of the original model.

