# OpenReview forum: "Sign and Basis Invariant Networks for Spectral Graph Representation Learning"
_NeurIPS.cc/2022/Conference — NeurIPS 2022 Submitted_

### Official Review · Reviewer_Lzrs · 2022-07-11

**Rating:** 6
**Confidence:** 3
**Soundness:** 3 good
**Presentation:** 3 good
**Contribution:** 3 good

**Summary:**

This paper proposes a SignNet and Basisnet to obtain invariant graph neural network by using eigenspaces. The paper provide significant improvements with the proposed method with several applications.

**Questions:**

What are the limitations of the proposed methods?


All experiments show improved performance compared to baseline methods. I am wondering if there are any data that may not give inferior performance. Are there any datasets or settings where the proposed method may not give good performance?


Is it possible to directly compare permutation based invariant/equivariant graph neural networks with the proposed method?


**Limitations:**

Limitation are not clearly stated.

**Strengths And Weaknesses:**

This paper makes an a useful and novel contribution to invaraint graph neural networks. Overall, the paper is well written and easy to understand. Experiments are extensive and covers several domains. All experiments show improved performances. However, the paper does not indicate any evidence where the proposed method could provide inferior performance.

---

> ### Author Response · Authors · 2022-08-02
> **Reply to Reviewer Lzrs**
>
> We thank the reviewer for stating that our paper makes a “useful and novel contribution to invaraint graph neural networks”, for noting that the paper is “well written and easy to understand”, and for noting that our experiments “are extensive” and “show improved performances.” In addition to the limitations we discussed in Section 7, we have added more limitations (see red text in Section 7 of the revised pdf). Below we address some points that the reviewer brought up; we hope that the reviewer may consider raising their score, please let us know if you have any further questions.
>
> > “​​Is it possible to directly compare permutation based invariant/equivariant graph neural networks with the proposed method?”
>
> If we have correctly understood the reviewer, we do already compare to baseline GNN models, which are inherently permutation invariant / equivariant. For instance, in Table 1 our SignNet significantly improves performance over the baseline GNN (which is the base model + No PE row). For instance, GatedGCN + No PE has an error of 0.252, while GatedGCN + SignNet PE has an error of 0.100. That is, **we reduce the error by 60% over the permutation invariant/equivariant GNN.**
>
> > “What are the limitations of the proposed methods? All experiments show improved performance compared to baseline methods. I am wondering if there are any data that may not give inferior performance. Are there any datasets or settings where the proposed method may not give good performance?”
>
> The molecular graph regression tasks that we considered seem to benefit from models that can capture structural properties of the graph. Thus, our models work well here, as the learned positional encodings from our SignNet can capture strong structural information. If the task does not require much processing of graph structure, such as if the node features are already very strong, then we expect our SignNet to not bring much benefit. For instance, node features are very strong in many common node classification tasks [Zhu et al. 2020], and some biological tasks on molecular graphs do not benefit much from positional encodings [Dwivedi et al. 2022].
>
> **References:**
> [Zhu et al. 2020] “Beyond Homophily in Graph Neural Networks: Current Limitations and Effective Designs”. NeurIPS 2020.
> [Dwivedi et al. 2022]. “Graph Neural Networks with Learnable Structural and Positional Representations”. ICLR 2022.

---

### Official Review · Reviewer_LMmf · 2022-07-11

**Rating:** 6
**Confidence:** 4
**Soundness:** 3 good
**Presentation:** 2 fair
**Contribution:** 3 good

**Summary:**

This paper proposes SignNet and BasisNet, two neural architectures, which can approximate any continuous function of eigenvectors with  invariances. The authors demonstrate theoretically that the proposed mothods  can approximate spectral graph convolutions and compute powerful spectral invariants. Experiments on several tasks illustrate the empirical benefits of the models’ approximation power and invariances.

**Questions:**

Should eigendecomposition be carried out if SignNet is to be utilized for GNNs, learning arbitrary Spectral Graph Convolutions? If so, does its high time cost limit its scalability?

**Limitations:**

Yes. The authors suggest that a potential limitation of the proposed method is that it only consider eigenspaces, not general subspaces.


**Strengths And Weaknesses:**

### Strengths:
1) The symmetries and  invariances of eigenspaces are significant and promising problems.
2) The advantages of using SignNet and BasisNet is clear,  making sign flips and choices of eigenspace bases invariant.
3) The theoretical analysis and proofs of the proposed method looks sound to me.
4) The experiments reveal some intriguing findings. SignNet, for example, encodes bilateral symmetry and structural information on a cat model, which seems to demonstrate its great performance.

### Weaknesses:
1) The technical section of the paper is relatively dense, which may be difficult for non-experts in the invariant Networks to understand.
2) The papers writing may need to double check. For instance, the references (1) and (2) in lines 92, 93, and 98 are invalid.

---

> ### Author Response · Authors · 2022-08-02
> **Reply to Reviewer LMmf**
>
> We thank the reviewer for noting that studying invariances of eigenspaces is “significant and promising”, for understanding the “clear” advantages of SignNet and BasisNet, for noting that our theory and proofs seem “sound”, and for noting that our experiments lead to some “intriguing findings” and “illustrate the empirical benefits” of our models. We individually address some other points raised here. We ask that the reviewer ask further questions if any arise, and that they consider raising their score if our changes are satisfying.
>
> > “The technical section of the paper is relatively dense, which may be difficult for non-experts in the invariant Networks to understand.”
>
> Thank you for this feedback. In our revision, we have moved the explanatory diagram of the positional encoding pipeline to the main paper. Further, we add more background and motivation for the problems we consider. Also, we have added some clarifying statements and further explanation to parts of the main paper. Please see the red text in that revised pdf for details, and let us know if you have any more comments on this.
>
> > “Should eigendecomposition be carried out if SignNet is to be utilized for GNNs, learning arbitrary Spectral Graph Convolutions? If so, does its high time cost limit its scalability?”
>
> We indeed need to precompute an eigendecomposition when using SignNet. However, this only needs to be done once before training, so the time cost is trivial For instance, it takes 15 seconds to do so for the ZINC dataset, and 330 seconds for the ZINC-full dataset (about 250k graphs). This is a negligible cost, and we need only do it once per dataset, after which we can train hundreds of different models as needed. We have added this to the main paper (red text in Section 4.1.).
>
> > “The papers writing may need to double check. For instance, the references (1) and (2) in lines 92, 93, and 98 are invalid.”
>
> Apologies, we changed these to (a) and (b) in-text; we see why these would be confused with the equation numberings. We have fixed this minor mistake, please let us know if you have any other specific comments on the writing.

---

> > ### Comment · Reviewer_LMmf · 2022-08-08
> > **Re**
> >
> > Thanks for the response and additional experiments! I have no more problem with this paper. Wish the authors the best of luck.

---

> > > ### Author Response · Authors · 2022-08-08
> > > **Second Reply to Reviewer LMmf**
> > >
> > > Great! We are glad that you have “no more problem with this paper”. **We kindly ask the reviewer to consider increasing their score**; the reviewer’s mentioned weaknesses were about some writing issues that we have addressed, and the reviewer had positive words for the importance of dealing with eigenspace symmetries, our theoretical analysis, and our experiments.
> > >
> > > Please let us know if you have any further questions or comments.

---

### Official Review · Reviewer_1KhG · 2022-07-11

**Rating:** 7
**Confidence:** 3
**Soundness:** 2 fair
**Presentation:** 3 good
**Contribution:** 3 good

**Summary:**

This work proposes a node position encoding scheme for graph representation learning, which respects two symmetric properties of an eigensystem, such as sign invariance, permutation invariance, and basis invariance. The new graph representation is believed spectral invariant and they are more capable for bio- or chemo-applications, where the graph substructures and global graph properties are important.

**Questions:**

1. Figure 1, why there are 0-vector rows/columns in the P matrix? (the reviewer assumes black boxes are 1 and the rest gray area are filled with 0 values.)
2. Line 91, it seems (2) defines basis invariant, why here is discussing permutation invariance instead?
3. Line 122, does any arbitrary GNNs satisfy permutation equivariant? If not, this statement is inaccurate.
4. Line 162 (corollary 1), why sign invariant is not discussed here? Does basis invariant always imply sign invariance?
5. Line 176, why the spectral convolution can be written in this form? Usually a spectral graph convolution is defined as $f(X) =V^{\top}\theta VX$. The result keeps the dimension of $n\times q$. However here $f(V,\Lambda,X)\in\R^{q}$ is a vector (the understanding might be incorrect here, but the reviewer assumes $\theta$ is a vector. Even though, unless $\theta\in\R^{n\times n}$, the new dimension is inconsistent with the former form.
6. Line 201, does it imply that spectral invariant equals basis invariant and permutation invariant?
7. In the experiment, why BasisNet is not studied?
8. Line 241, is $k$ a hyper-parameter?
9. Figure 5 (it is not a question though), consider changing orange to another color, as both node feature and position encoding modules are using the orange color, but they are not directly connected.


**Limitations:**

Yes

**Strengths And Weaknesses:**

strengths:
- originality: The authors value the important spectral information provided by eigenvectors of the graph Laplacian, which are usually ignored by message passing methods. On top of that, they make transformations to the eigenvectors to encode position information of a graph while preserving certain properties. Such symmetric properties, to the best of the reviewer’s knowledge, have never been analyzed by existing work.

- quality: Abundant analyses were provided in the main text and in appendices to consolidate the encoding scheme. In addition, experiments are designed from different perspectives to support the main claims of the work.

weaknesses:
- clarity: While the reviewer acknowledges that there are too many details to show in a limited number of pages, introducing the main algorithm in the appendix indeed prevents a reader from immediately understanding the workflow of the proposed method in a representation learning task. For the general audience, especially those without a solid mathematical background, to build a holistic view effortlessly, the reviewer highly recommends that the authors could consider rearranging some parts of the paper. In addition, some paradigms are used before a proper definition, which makes the paper more difficult to follow. For instance, the “higher dimensional basis” in the introduction was discussed without first explaining what is it.

- significance: The reviewer was mainly concerned with the empirical significance of the framework. Although it is claimed that symmetries are important for an eigensystem, and capturing graph properties (such as substructures) are important for bio- or chemo-applications, the experiments did not show a significant underperformance of the baseline methods over the proposed SignNet. A simple demonstration showing how ignoring such properties cause trouble in graph representation learning would be helpful. In addition, as the model does not outperform other methods significantly on the three benchmark datasets, it is questionable how much more computational cost was made to trade for a smaller MAE score?

---

> ### Author Response · Authors · 2022-08-02
> **Reply to Reviewer 1KhG (Part 1/2)**
>
> We thank the reviewer for noting the “originality” of our work and praising the “abundant analyses”. We also thank the reviewer for their notes on clarity and numerous questions; we have integrated changes (see red text in our revised pdf) based on the much-appreciated close reading done by the reviewer. However, we disagree on the statements about the significance of our empirical results, and hope that the reviewer will increase their score after reading our rebuttal.
>
>
> ## **Weakness: empirical performance**
> > “the experiments did not show a significant underperformance of the baseline methods over the proposed SignNet”
>
> We strongly disagree with this statement by the reviewer. For one, the 3 other reviewers (xhbi, LMmf, Lzrs) have positive words for our experiments, noting that our experiments are “extensive and covers several domains” (Lzrs), and that they “illustrate the empirical benefits” of our models (LMmf). Here we give more details on why our models show empirical benefits: we clearly and consistently beat baselines that use Laplacian eigenvectors, and we are competitive with and beat domain-agnostic SOTA while not adding much computational cost.
>
> > “A simple demonstration showing how ignoring such properties [symmetries of eigensystems and capturing graph properties] cause trouble in graph representation learning would be helpful”
>
> Perhaps we did not emphasize this enough in the original draft, but this is indeed present in our paper; we will make this clearer in the revision. Table 1 shows **extremely large benefits when using our SignNet to respect symmetries of eigenvectors:**
>
> 1. SignNet is consistently better by a very large margin over the base models without eigenvectors (the “No PE” rows), as it reduces error by over 60% when using Sparse Transformer or GatedGCN as the base model.
> 2. SignNet reduces error by about 50% compared to all eigenvector baselines for the Sparse Transformer and GatedGCN, and by over 33% for the other base models. These baselines that use Laplacian eigenvectors with heuristics for sign invariance (sign flipping, absolute value, sign canonicalization) are common methods from the prior literature, and our approach is both much more principled and empirically stronger.
> 3. SignNet is similarly much better than eigenvector baselines and No PE methods on counting substructures (Figure 2) and texture reconstruction (Table 3).
>
> > “as the model does not outperform other methods significantly on the three benchmark datasets, it is questionable how much more computational cost was made to trade for a smaller MAE score?”
>
> The computational cost of our methods are slightly higher than that of standard message passing GNNs, but **much lower than many other listed methods**, and we significantly outperform standard message passing. For example, GatedGCN with no eigenvectors takes about 8.2 seconds per training iteration on ZINC, and GatedGCN with 8 eigenvectors and SignNet takes about 10.6 seconds; this is only a 29% increase in time, for a reduction of test MAE by over 50% with SignNet! In contrast, CIN [Bodnar et al. 2021] has a 136% increase in training epoch time over the GIN baseline (Table 5 in [Bodnar et al. 2021]). Furthermore, $\delta$-2-GNN, SpeqNet, GNN-IR, and PF-GNN are generally several times slower than a comparable base message passing model (see Table 3 in [Morris et al. 2020b], Table 6+7 in [Morris et al. 2022], Table 5 in [Dupty and Lee. 2022], and Figure 3 in [Dupty et al. 2021]). We have added our runtime numbers to the paper, see red text in Section 5.1.
>
> Also, we disagree that our model “does not outperform other methods significantly” on the molecular graph regression tasks; note that our domain-agnostic SignNet significantly outperforms all domain-agnostic methods on ZINC and ZINC-full (see line-271), and we even outperform two domain-specific methods (HIMP and CIN-small). The domain-specific method CIN is within a standard deviation, which is remarkable, since our SignNet does not have domain information about molecules built into it.
>
> **References:**
> [Bodnar et al. 2021] “Weisfeiler and Lehman Go Cellular: CW Networks”. NeurIPS 2021.
> [Morris et al. 2020b] “Weisfeiler and Leman go sparse: Towards scalable higher-order graph embeddings”. NeurIPS 2020.
> [Morris et al. 2022] “SpeqNets: Sparsity-aware Permutation-equivariant Graph Networks”. ICML 2022.
> [Dupty and Lee. 2022] “Graph Representation Learning with Individualization and Refinement”. arXiv 2022.
> [Dupty et al. 2021] “PF-GNN: Differentiable Particle Filtering Based Approximation of Universal Graph Representations''. ICLR 2021.

---

> > ### Author Response · Authors · 2022-08-02
> > **Reply to Reviewer 1KhG (Part 2/2)**
> >
> > ## **Weakness: clarity**
> >
> > > “For the general audience, especially those without a solid mathematical background, to build a holistic view effortlessly, the reviewer highly recommends that the authors could consider rearranging some parts of the paper. “
> >
> > Thank you, this is valuable feedback that we have incorporated into our revision. As you point out, doing so will make it much easier for readers not familiar with some of the technical tools used to understand our method.  Specifically we have changed the following (see red text in the revised pdf):
> > * Moved the diagram (now called Figure 2) that explains how positional encodings are used to the main paper, and added to the caption. This will be a key clarification for readers previously unfamiliar with positional encodings.
> > * We added more content that motivates the sign and basis invariance problem specifically in graph positional encoding (see red text in beginning of Section 2), and other background.
> > * We have added some other clarifying statements and further explanation to parts of the main paper. Please see the revision for details.
> >
> > ### Questions
> >
> > > “Figure 1, why there are 0-vector rows/columns in the P matrix? (the reviewer assumes black boxes are 1 and the rest gray area are filled with 0 values.”
> >
> > We have added nonzeros to the rest of the rows / columns, likely, some black squares were lost when we inserted the text “P”.
> >
> > > “Line 91, it seems (2) defines basis invariant, why here is discussing permutation invariance instead?”
> >
> > We accidentally used (2) to enumerate a point, when it is also already an equation numbering. We have changed the enumeration to (a), (b) instead for clarity, thanks!
> >
> > > “Line 122, does any arbitrary GNNs satisfy permutation equivariant? If not, this statement is inaccurate.”
> >
> > Standard GNNs do. We will change the phrasing to “most standard GNNs”.
> >
> > > “Line 162 (corollary 1), why sign invariant is not discussed here? Does basis invariant always imply sign invariance?”
> >
> > Yes, basis invariance implies sign invariance, as diagonal matrices of the form $\mathrm{Diag}(\pm 1, \ldots, \pm 1)$ are orthogonal. We note this in line 74, and in the revision we have made this more clear in the context of BasisNet and corollary 1.
> >
> > > “Line 176, why the spectral convolution can be written in this form?”
> >
> > The output of a spectral graph convolution (as we define it) is indeed in $\mathbb{R}^{n \times q}$ for an input $X \in \mathbb{R}^{n \times q}$. For a vector $\theta \in \mathbb{R}^n$, we can write a spectral graph convolution as $V \mathrm{Diag}(\theta) V^\top X = \sum_i \theta_i v_i v_i^\top X$, where $\mathrm{Diag}(\theta) \in \mathbb{R}^{n \times n}$ is the diagonal matrix with $\theta$ on the diagonal. We have written out this matrix form ($V\mathrm{Diag}(\theta) V^\top X$) in the paper to make it clearer.
> >
> > > “Line 201, does it imply that spectral invariant equals basis invariant and permutation invariant?”
> >
> > Yes, a spectral invariant must be basis invariant and permutation invariant when viewed as a function on eigenvectors / eigenvalues. This is a requirement for the spectral invariant to be a well-defined function.
> >
> > > “In the experiment, why BasisNet is not studied?”
> >
> > We do use BasisNet in Table 8 of the Appendix. For other experiments like molecular graph regression, we compare against the recent literature, which mostly just uses sign flipping data augmentation for sign invariance; we show that sign invariance alone with SignNet gives a lot of benefit.
> >
> > > “Line 241, is k a hyper-parameter?”
> >
> > Correct, as in many other works we choose some $k$ and take the bottom $k$ Laplacian eigenvectors of each graph. When we use SignNet with all eigenvectors of all graphs, we need not choose $k$.
> >
> > > “Figure 5 (it is not a question though), consider changing orange to another color ...”
> >
> > Thanks for pointing this out, we have changed the color for clarity.

---

> > > ### Comment · Reviewer_1KhG · 2022-08-08
> > > **reply to authors response**
> > >
> > > Thanks for the very detailed response to the questions or the concerns. The revised submission is clearer and easier to follow. There are two remaining questions/comments:
> > >
> > > 1. Empirical performance: the comment was made mainly from the results in Tables 2&3. As the highlighted scores are not always from SignNet, the reviewer does not think it is fair to say that the proposed method “clearly and consistently beat baselines”. (For clarity, it does NOT imply that the proposed model has to significantly outperform all existing methods to be a novel one. The reviewer only disagrees with the aggressive statement of “clearly and consistently beat baselines”.)
> > >
> > > 2. Questions-"line 122, permutation equivariant GNN": what is considered a “standard” GNN? (By the way, the phrase has not been changed, in line 101 of the revised version)

---

> > > > ### Author Response · Authors · 2022-08-08
> > > > **Second Reply to Reviewer 1KhG**
> > > >
> > > > Thank you for the reply. We are glad that you find the revised version to be “clearer and easier to follow”, we think that the reviewers’ suggestions have helped the writing of our paper! We reply to your remaining questions / comments below.
> > > >
> > > > Given that we have made these improvements to the exposition of the paper, as suggested by the reviewer, **we kindly ask that the reviewer consider increasing their score**. If there are any further questions / comments, feel free to reply to us.
> > > >
> > > > > “Empirical performance: … As the highlighted scores are not always from SignNet, the reviewer does not think it is fair to say that the proposed method “clearly and consistently beat baselines”. ”
> > > >
> > > > Thank you for clarifying the point that you are making. We agree that we should not have given such a strong statement about empirical improvement in our short rebuttal, and we refrain from making such statements in our paper. In the paper we make more qualified, specific statements comparing our method to others.
> > > >
> > > > For instance, when describing the results in Table 2, we say:
> > > >
> > > > “SignNet outperforms all domain-agnostic methods on ZINC and ZINC-full, and is within a standard deviation of the best domain-specific method. Our mean score is the second best on Alchemy, and is within a standard deviation of the best. We perform much better on ZINC (.084) than other state-of-the-art positional encoding methods, like GNN-LSPE (.090)~[Dwivedi et al. 2022], SAN (.139) [Kreuzer et al. 2021], and Graphormer (.122) [Ying et al. 2021].”
> > > >
> > > > And when describing the results in Table 3, we say: “We see that the SignNet architecture improves over the original Intrinsic NF model and over other baselines — especially in the LPIPS (Learned Perceptual Image Patch Similarity) metric”
> > > >
> > > > > “Questions-"line 122, permutation equivariant GNN": what is considered a “standard” GNN? (By the way, the phrase has not been changed, in line 101 of the revised version)”
> > > >
> > > > Thank you for catching that, we will also change line 101. We will change our sentence to “most standard GNNs — such as all message passing GNNs [Gilmer et al. 2017], which include GCN [Kipf and Welling. 2017], GIN [Xu et al. 2019], and GAT [Veličković et al. 2018]“.

---

> > > > > ### Comment · Reviewer_1KhG · 2022-08-09
> > > > > **final comments**
> > > > >
> > > > > Thanks to the authors for their detailed and patient response to all my concerns and questions. The explanation helped in clarifying my concerns. I have therefore decided to change my score to 7.

---

### Official Review · Reviewer_xhbi · 2022-07-11

**Rating:** 3
**Confidence:** 4
**Soundness:** 2 fair
**Presentation:** 3 good
**Contribution:** 2 fair

**Summary:**

Contributions:
(1) A neural architecture that preserves two symmetries (i) sign invariance and (ii) basis invariance, built based on the existing equivariant neural networks.
(2) Extensive experiments on various settings.

**Questions:**

I don't see the theoretical value of Theorem 2.

A quick review of spectral graph convolution: The spectral graph convolution is built based on Graph Fourier Transform (GFT) as in [Bruna et al., 2014] and [Defferrard et al., 2016]. GFT constructs the Fourier basis as the eigenvectors of the (normalized) graph Laplacian. GFT allows us to do Fourier transform of a graph signal or function f: V -> R where V is the set of nodes (i.e. each node is associated with a number) into the spectral domain. Then, based on the convolution theorem, we can define the graph convolution between a graph signal and a filter based on their Fourier transforms in the spectral domain.

In this paper, given the construction of SignNet and BasisNet using the eigenvectors, I think it is just a tweaked form of spectral graph convolution. The authors use the existing permutation equivariant neural networks and add the minus sign to the eigenvectors, but I find this modification brings no novelty.

**Limitations:**

The reasons we want a neural network to preserve symmetries and build equivariant neural networks are because these symmetries arise in the data. In graph and molecule data, only two symmetries are important: permutation and rotation. We want permutation- and rotation-equivariant neural networks to learn and generate graph and molecule data, because both theoretical and empirical evidences have shown that preserving the two symmetries reduces the amount of training data, makes the model robust against data noise and smaller (i.e. less parameters), and importantly allows us to embed physical knowledge into the model. These are the fundamental motivations for preserving permutation and rotation symmetries that obviously arise from the data.

The fundamental limitation of this work is a lack of motivation and evidence for the two symmetries they propose to preserve (i) sign invariance and (ii) basis invariance. I highly doubt if these two symmetries are relevant and important. Given this limitation, I vote to reject this paper. However, I keep the rating to be 4 (borderline reject) because I am willing to listen to the authors' rebuttals.

**Strengths And Weaknesses:**

Strengths:
(1) Paper is easy to follow and well-written.
(2) The proposed methods are simple to implement using the existing codes of permutation equivariant neural networks.
(3) Many experiments.

Weaknesses: It is not clear "why" to preserve the two symmetries this paper proposes to preserve including sign invariance and basis invariance. In graph & molecule setting, two important symmetries are permutation (i.e. if we permute the nodes/atoms, the output stays the same) and rotation (i.e. if we rotate the 3D coordinates associating with each atom, the prediction for a physical property stays the same). I don't see a strong evidence that suggests preserving sign & basis invariances on top of permutation & rotation invariances brings any benefits.

---

> ### Author Response · Authors · 2022-08-02
> **Reply to Reviewer xhbi (Part 1/2)**
>
> Thank you for taking time to review our work. We are glad you found the paper clear, with many experiments, and providing an easy to implement method. Your review noted some concerns, each of which we address below. Your main concern—that real data (such as molecules) do not have the symmetries we study—arises from a key confusion about how SignNet is used. We explain the details below, and believe that this clarification should lead to a significantly revised understanding of our contribution. We have also updated our pdf — if you have time, **please see Figure 2 and its updated caption in our revised pdf** (previously in the Appendix). Also, we have added a section “Graph Positional Encodings” in red text at the beginning of Section 2, which summarizes some of the points we make here.
>
> We thank the reviewer for being “willing to listen to the authors’ rebuttals”, and we hope that the reviewer may consider raising their score; please let us know if you have any further questions.
>
> ## **Why sign and basis invariance?**
> > “It is not clear "why" to preserve the two symmetries this paper proposes to preserve including sign invariance and basis invariance.”
> > “In graph and molecule data, only two symmetries are important: permutation and rotation …”
>
> We give multiple points here to help clarify the motivation and benefits of sign / basis invariance.
>
> 1. **What we are doing**: SignNet and BasisNet are _not_ applied to input graph data such as molecules in the usual way a GNN is used. Instead our networks take as input _eigenvectors_, which are used to learn _positional encodings_. Positional encodings are additional features that are concatenated to the existing node features, and describe the position of each node in the graph; these have been found to improve performance in many graph learning tasks. Figure 2 (originally in the Appendix) shows the pipeline clearly. We have moved this figure to the main text to improve reliability and ensure this point is clear to all future readers.
> 2. **Precedent in the literature**: A rapidly growing literature aims to make more powerful GNNs / Graph Transformers by using positional encodings, and one of the most popular positional encodings are the eigenvectors of the graph Laplacian. When using these eigenvectors, **many papers have _directly stated_ the desire for sign invariance or basis invariance**; in other words, **this is a well established problem in the literature**, see for instance in the context of graph positional encodings: [Kreuzer et al. 2021] page 5, [Beaini et al. 2021] page 6, [Mialon et al. 2021] page 5, [Dwivedi and Bresson 2021] page 3, and [Dwivedi et al. 2022] page 3. We make the first neural models that take eigenvectors as input and solve these problems noted by so many works.
>
>     In fact, the sign and basis invariance problems have been mentioned in works from many different fields, such as when processing eigenfunctions of the Laplace-Beltrami operator on manifolds [Rustamov. 2007] page 4. As suggested by our texture reconstruction experiments in Section 5.3, SignNet is also promising for applications in these other fields.
> 3. **Why sign / basis invariance**: The choice of signs in the eigenvectors returned by an eigensolver are random. Likewise, basis changes in eigenspaces of higher dimension are random. These are artifacts of the solver, and have absolutely no effect on your desired labels in a graph learning task. When we add eigenvectors to a prediction model for a graph task, we hence want invariance to these artifacts, to not be sensitive to this random noise that is not relevant to our task.
> 4. **An Analogy:** Just like models that do not use 3D coordinates of atoms do not have rotational invariance built in, models that do not use eigenvectors do not have sign invariance built in. Since we propose the first neural models that take in eigenvectors and have sign invariance, we understand why the reviewer may not see why this is needed. Analogously, if all molecular models were based on 2D molecular graphs, and the very first rotation invariant model was presented, a reader may not understand why rotation invariance is desired.
>
> **References:**
> [Kreuzer et al. 2021] “Rethinking Graph Transformers with Spectral Attention”. NeurIPS 2021.
> [Beaini et al. 2021] “Directional Graph Networks”. ICML 2021.
> [Mialon et al. 2021] “GraphiT: Encoding Graph Structure in Transformers”. arXiv 2021.
> [Dwivedi and Bresson 2021] “A Generalization of Transformer Networks to Graphs”. DLG-AAAI 2021.
> [Dwivedi et al. 2022] “Graph Neural Networks with Learnable Structural and Positional Representations”. ICLR 2022.
> [Rustamov. 2007] “ Laplace-Beltrami Eigenfunctions for Deformation Invariant Shape Representation”. Symposium on Geometry Processing 2007.

---

> > ### Author Response · Authors · 2022-08-02
> > **Reply to Reviewer xhbi (Part 2/2)**
> >
> > > “ I don't see a strong evidence that suggests preserving sign & basis invariances on top of permutation & rotation invariances brings any benefits.”
> >
> > We hope that the motivation and arguments detailed above offer sufficient motivation that sign and basis invariance can be beneficial. In addition, the paper shows empirical evidence:
> > * **In Table 1**, results in the rows “LapPE (flip)” use eigenvectors in standard GNNs without exact sign invariance, whereas rows “SignNet” use eigenvectors in GNNs with our SignNet giving sign invariance. We see massive performance gains when using SignNet to give sign invariance: e.g. GatedGCN, the Test MAE goes from .198 without sign invariance to .100 with sign invariance.
> > * **In Figure 3**, the blue bars “LapPE” once again use eigenvectors in standard GNNs without exact sign invariance, whereas the red bars “SignNet” have exact sign invariance. Again we see large benefits from sign invariance.
> > * **In Table 3**, the rows “Intrinsic NF” and “Sign flip” do not have exact sign invariance, and they are once again outperformed by our SignNet.
> >
> >
> > ## **Relation to Spectral Graph Convolution**
> > > “given the construction of SignNet and BasisNet using the eigenvectors, I think it is just a tweaked form of spectral graph convolution. The authors use the existing permutation equivariant neural networks and add the minus sign to the eigenvectors, but I find this modification brings no novelty.”
> >
> > We strongly disagree with these statements; SignNet and BasisNet are not just spectral graph convolutions; we justify this in several ways. Intuitively, the spectral graph convolution is forced to directly multiply eigenvectors against a node signal matrix $V\mathrm{Diag}(\theta) V^\top X$, while SignNet and BasisNet can process eigenvectors nonlinearly, can learn to ignore node features, can learn to ignore spectral information, can learn independent functions of eigenvectors and node features, and can multiply nonlinear functions of the eigenvectors, such as $\varphi(V) \mathrm{Diag}(\theta) \varphi(V)^\top X$ for some nonlinear $\varphi$. We further give rigorous theoretical justification showing that SignNet and BasisNet _provably_ go beyond spectral graph convolutions:
> >
> > **Theoretical justification.** We show that **SignNet and BasisNet are provably stronger than spectral graph convolutions.** Recall that Theorem 2 shows SignNet and BasisNet can approximate any spectral graph convolutions. In these following results, which we have added to the paper, we show that the converse is not true: spectral graph convolutions **cannot**  approximate SignNet or BasisNet in general. As this general function approximation perspective may be less familiar, we use the familiar example of graph isomorphism testing to illustrate the following.
> >
> > **Proposition (See Prop 3 in revision):** There exist infinitely many non-isomorphic graphs that spectral graph convolutions or spectral GNNs cannot distinguish, but SignNet and BasisNet can distinguish.
> >
> > The proof is in the Appendix G.1 in our paper revision. Note that this result also applies to spectral GNNs, with elementwise nonlinearities and arbitrary number of layers.
> >
> > As another set of examples, the strength of spectral graph convolutions is known to be limited in several ways that our SignNet and BasisNet are not — see the recent work [Wang and Zhang 2022]. For instance, spectral GNNs with polynomial filters are bounded in power by the 1-WL test (Prop 4.3 in [Wang and Zhang  2022]), while our SignNet and BasisNet go beyond 1-WL (see e.g. discussion in our Section 4.2). Further, standard spectral graph convolutions only apply the same single filter to each feature channel (that is, they are of the form $p(L) X$ for $X$ in $R^{n \times q}$, see Proposition 4.2 in [Wang and Zhang 2022]), whereas our SignNet and BasisNet can learn to apply different filters to each feature channel (that is, we can learn functions of the form $[p_1(L)X_1, …, p_q(L)X_q]$, where $X_i$ is the $i$th column of $X$); this follows from a trivial extension to the proof of Theorem 2.
> >
> >
> > > “I don't see the theoretical value of Theorem 2.”
> >
> > We hope that we made the theoretical value of Theorem 2 clearer with this above explanation; SignNet and BasisNet are quite different from spectral graph convolutions. In fact, they can simulate spectral graph convolutions, and approximate other functions that spectral convolutions cannot express.
> >
> > **References:**
> > [Wang and Zhang 2022] “How Powerful are Spectral Graph Neural Networks”. ICML 2022.

---

### Author Response · Authors · 2022-08-02
**General comment to reviewers**

We would like to thank everyone involved in the review process for their much-appreciated work. We have **revised our PDF, please see the red text for new additions**. We added a little less than a page of new content to the main paper (as is allowed in the 10 page camera-ready), but to stay in the 9 page limit we temporarily moved the decomposition theorem / universality section (now Appendix A) and one of the visualizations of SignNet output. We will move this content back into the main paper for the camera-ready. Also, we added more content to the Appendix, including a proof of a new theoretical result. In particular, we added:

* **More positional encoding background:** Moved diagram of positional encoding pipeline (now Figure 2) to the main text, and added to the caption. Also added a paragraph to the start of Section 2 that gives more background on sign and basis invariance in graph positional encodings. This should help clarify the basic background and learning framework that we use.
* **More spectral graph convolution theory:** Added a new theoretical result (Proposition 3) that more clearly shows the power of SignNet and BasisNet over spectral graph convolution: SignNet and BasisNet can distinguish infinitely many pairs of non-isomorphic graphs that spectral graph convolutions and spectral GNNs cannot distinguish.
* **Runtime numbers:** We add specific runtime numbers to show that our method and the computation of eigenvectors are efficient, see Section 4.1.
* **Clarity:** Expanded some acronyms to help with readability
* **Clarity:** Added function domains and ranges in several places (e.g. $f: \mathbb{R}^{n \times d} \to \mathbb{R}^{n \times d}$) for clarity.
* **Clarity:** Fixed small writing issues noted by reviewers.

---

### Meta-Review · Area_Chair_Kw3A · 2022-08-30

**Recommendation:** Reject
**Confidence:** Less certain

**Metareview:**

This paper proposes new neural architectures that are invariant to sign and basis, named SignNet and BasisNet. Some reviewers raised the question of why such invariances are needed, and they noticed that only permutation and rotation symmetry arise in all the data mentioned in the paper. One of the reviewers strongly believes that the proposed approach is a simple tweak from the prior work "Invariant and Equivariant Graph Networks" (IGN) and the complexity of the proposed method is much higher than the spectral model using the eigenvectors. In their response, the authors provided some explanation and theoretical analysis on why SignNet and BasisNet are different from spectral graph convolutions. They argue that they can simulate spectral graph convolutions, and approximate other functions that spectral convolutions cannot express.

**Award:**

No

---

### Decision · Program_Chairs · 2022-09-14

Reject